# Deciphering the exact breakpoints of structural variations using long sequencing reads with DeBreak

Yu Chen ⊕[1,2], Amy Y. Wang ⊕[2,3], Courtney A. Barkley ⊕[1], Yixin Zhang[4], Xinyang Zhao[5], Min Gao[2,6], Mick D. Edmonds[1] & Zechen Chong ⊕[1,2,7] ✉

Long-read sequencing has demonstrated great potential for characterizing all types of structural variations (SVs). However, existing algorithms have insufficient sensitivity and precision. To address these limitations, we present DeBreak, a computational method for comprehensive and accurate SV discovery. Based on alignment results, DeBreak employs a density-based approach for clustering SV candidates together with a local de novo assembly approach for reconstructing long insertions. A partial order alignment algorithm ensures precise SV breakpoints with single base-pair resolution, and a k-means clustering method can report multi-allele SV events. DeBreak outperforms existing tools on both simulated and real long-read sequencing data from both PacBio and Nanopore platforms. An important application of DeBreak is analyzing cancer genomes for potentially tumor-driving SVs. DeBreak can also be used for supplementing whole-genome assembly-based SV discovery.

Structural variations (SVs), or genomic rearrangements, including insertions, deletions, inversions, duplications, translocations, and complex forms of multiple events, contribute a large proportion of genetic variations in many species. In humans, SVs affect larger genomic region size than any other type of variants[1–4] and play a pathogenic role in a wide range of genetic disorders[5–9]. SVs are also associated with diverse phenotypes in non-human organisms[10–12]. Therefore, comprehensive characterization of all forms of SVs is critical for fully understanding their contribution to genetic diversity, species divergence, and other phenotypic traits.

The currently available real-time long-read sequencing platforms, Pacific BioSciences (PacBio) and Oxford Nanopore, generate very long reads (>20 kbp on average) and have demonstrated superior performance over short reads on SV discovery. For example, many rare genetic diseases have been solved using long-read sequencing technologies[13–16]. Long-read sequencing can potentially delineate the full landscape of SVs in individual genomes. By sequencing and analyzing a haploid human genome (CHM1) using single-molecule, real-time (SMRT) DNA sequencing, Chaisson et al.[17] resolved the complete sequence of 26,079 euchromatic structural variations, which is a six-fold increase when compared with prior work[2] using short read sequencing. Recent work by the Human Genome Structural Variation Consortium (HGSVC) resulted in a sevenfold increase in SV number using multiple platforms in which the PacBio results contributed the most[3].

Although great strides have been made, existing computational tools for SV detection using long reads remain few in number and can be further enhanced and optimized. These methods usually can only

[1]Department of Genetics, Heersink School of Medicine, University of Alabama at Birmingham, Birmingham, AL 35294, USA. [2]Informatics Institute, Heersink School of Medicine, University of Alabama at Birmingham, Birmingham, AL 35294, USA. [3]Department of Medicine, Division of General Internal Medicine, Heersink School of Medicine, University of Alabama at Birmingham, Birmingham, AL 35294, USA. [4]Department of Computer Science, College of Arts and Sciences, University of Alabama at Birmingham, Birmingham, AL 35294, USA. [5]Department of Biochemistry and Molecular Genetics, Heersink School of Medicine, University of Alabama at Birmingham, Birmingham, AL 35294, USA. [6]Department of Medicine, Division of Cardiovascular Disease, Heersink School of Medicine, University of Alabama at Birmingham, AL 35233 Birmingham, USA. [7]HudsonAlpha Institute for Biotechnology, Huntsville, AL 35806, USA. ✉e-mail: zchong@uabmc.edu

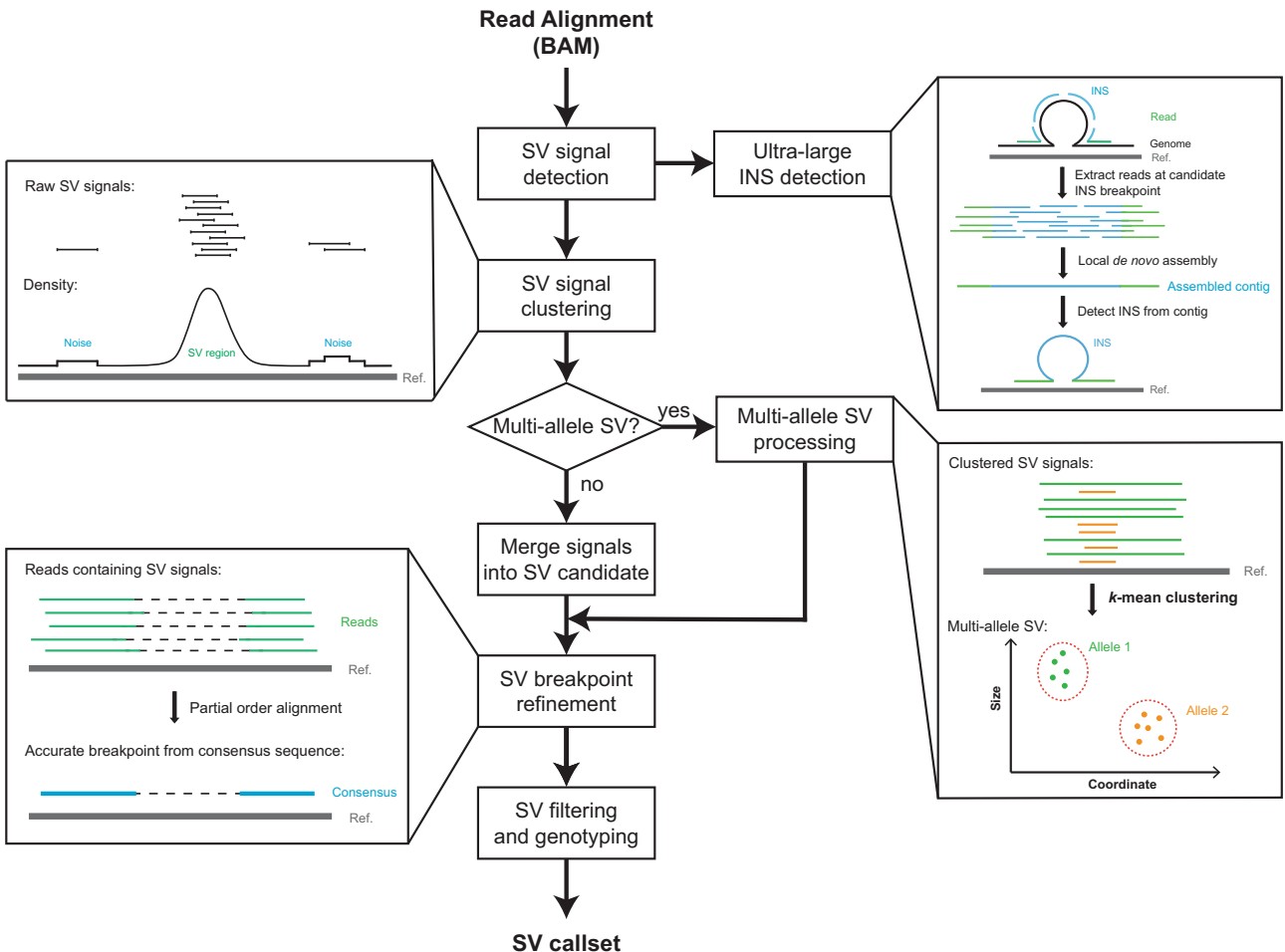

**Fig. 1 | Workflow of DeBreak.** The major steps of DeBreak SV discovery include SV signal detection, signal clustering, breakpoint refinement, and filtering and genotyping. Detailed descriptions of each step can be found in the "Methods" section.

characterize a subset of SVs, and sensitivity and precision are not ideal. For example, PBHoney[18] uses BLASR[19] to map the PacBio subreads to collect soft-clipped reads and remap clipped tails (>200 bp) to compose a "piece alignment". However, PBHoney can only infer simple deletions, tandem duplications, inversions, and translocations and does not perform well for insertions and other types of SVs. Another alignment-based method, Sniffles[20] uses both within-read alignments and split-read alignments from the NGMLR aligner. Sniffles can analyze both PacBio and Nanopore sequencing data and report some complex forms of SVs besides simple SVs. For both PBHoney and Sniffles, the breakpoints inferred from clusters of alignments usually are not precise, preventing experimental validation and mechanism analysis[21–24] that rely on SV junction sequences. Moreover, both tools are deficient in detecting the full spectrum of SVs. For instance, long insertions close to or longer than nearby read lengths are often missed. These issues remain a concern with a recently published alignment-based method, CuteSV[25].

Besides alignment-based methods, methods using local de novo assembly of mapped reads are also applied to SV discovery[3,17,26]. These local assembly-based methods utilize reads from haploid genomes or phased reads from diploid genomes and perform de novo assembly in a window. The consensus sequences generated can usually identify precise breakpoints. However, they only report insertions, deletions, and subsets of inversions. During phasing, only about two-thirds of reads in each sample can be haplotype-partitioned. These partitions require different tools applied to the data from other platforms, preventing more generic and broader applications of SV analysis.

Whole genome de novo assembly can be considered the ultimate solution for SV characterization. Although researchers have made progress on this front[27–30], whole genome de novo assembly with long noisy reads is inherently challenging. It requires high-coverage sequencing data, and difficulties remain in dealing with long repetitive sequences, tandem repeats, as well as heterozygosity. Moreover, whole genome de novo assembly usually requires high-memory computing nodes and long running time, and it is difficult to evaluate accuracy.

Here, we present DeBreak (Deciphering Exact BREAKpoints), an algorithm for comprehensive and accurate SV discovery from long reads. DeBreak detects SV events using two different strategies, depending on whether SVs can be spanned within reads (Fig. 1, see the "Methods" section). For SVs contained within reads, DeBreak scans all read alignments for raw SV signals for each category of SV (Fig. S1) and then clusters these signals using a unique density-based clustering algorithm with flexible clustering window sizes (Fig. S2). This approach allows for accurate SV candidate identification for SVs with varying lengths and local sequence contents. In the next step, the SV breakpoint refinement with a partial order alignment (POA)[31] algorithm can accurately infer SV breakpoints with a single base-pair resolution. With its automatic sequencing depth estimation and parameter optimization, DeBreak filters SV candidates and reports a high-confidence SV callset with genotyping information. For SVs that are too large to be spanned within reads, DeBreak first identifies candidate SV breakpoints and then performs local de novo assembly to reconstruct SV-containing sequences (Fig. S3). DeBreak completes the analysis by

**Table 1 | SV discovery accuracy on simulated datasets**

| Type | DeBreak | | | Sniffles | | | pbsv | | | cuteSV | | |
|---|---|---|---|---|---|---|---|---|---|---|---|---|
| | Rec | Pre | F1 | Rec | Pre | F1 | Rec | Pre | F1 | Rec | Pre | F1 |
| **PacBio** | | | | | | | | | | | | |
| DEL | **99.59** | 99.50 | **99.54** | 95.50 | **99.83** | 97.62 | 97.21 | 99.58 | 98.38 | 96.94 | 99.82 | 98.36 |
| INS | **98.51** | **99.65** | **99.08** | 92.35 | 99.52 | 95.80 | 95.38 | 96.22 | 95.80 | 94.58 | 94.71 | 94.64 |
| DUP | **98.27** | 97.04 | **97.65** | 90.80 | 98.63 | 94.55 | 44.53 | 97.11 | 60.97 | 44.13 | **99.03** | 61.05 |
| INV | **99.10** | 99.40 | **99.25** | 94.57 | 97.22 | 95.88 | 82.67 | **99.92** | 90.47 | 96.53 | 50.00 | 65.88 |
| TRA | **99.17** | **99.67** | **99.41** | 97.67 | 95.60 | 96.62 | 97.67 | 25.04 | 39.86 | 99.00 | 50.30 | 66.70 |
| Total | **99.02** | 99.45 | **99.23** | 93.85 | **99.47** | 96.58 | 93.36 | 95.34 | 94.34 | 93.50 | 92.39 | 92.94 |
| **Nanopore** | | | | | | | | | | | | |
| DEL | 98.08 | **98.82** | 98.45 | 94.83 | 98.72 | 96.74 | **98.65** | 98.55 | **98.60** | 97.87 | 98.27 | 98.07 |
| INS | **98.17** | 99.68 | **98.92** | 92.70 | **99.73** | 96.09 | 97.05 | 95.92 | 96.48 | 96.04 | 94.01 | 95.01 |
| DUP | **97.00** | 93.60 | **95.27** | 91.97 | 98.37 | 95.06 | 46.03 | 87.37 | 60.23 | 39.67 | **99.50** | 56.72 |
| INV | 91.63 | 98.71 | **95.04** | 93.67 | 95.18 | 94.41 | 88.20 | **99.96** | 93.71 | **97.37** | 50.00 | 66.07 |
| TRA | 92.83 | **99.83** | 96.17 | 97.17 | 91.38 | 94.18 | 97.83 | 25.26 | 40.15 | **99.17** | 50.17 | 66.63 |
| Total | **97.73** | 98.96 | **98.35** | 93.71 | 98.91 | 96.24 | 95.08 | 94.60 | 94.84 | 94.41 | 91.46 | 92.91 |

The unit for recall, precision, and F1 score is %. The highest recall, precision, and F1 score among four tested SV callers are marked in bold.
*Rec* Recall, *Pre* Precision, *F1* F1 score.

integrating all identified SV events together to form a final, high-confidence SV callset.

## Results

### Benchmark on simulated dataset

To benchmark the performance of SV discovery, we first compared DeBreak with three SV callers, Sniffles, pbsv, and cuteSV, using in silico datasets. A total of 22,200 SVs were simulated and embedded into the human reference genome (GRCh38), serving as the ground truth. The sizes of simulated SVs follow similar distributions to those observed in real human samples, with Alu and LINE peaks[32] (Fig. S4a). PacBio-like and Nanopore-like reads were simulated based on a modified genome with pbsim[33] and Badread[34] and aligned to the human reference genome. To mimic long reads generated from different library preparation protocols, simulations were performed using three datasets with different insert sizes (Fig. S4b). We applied these SV callers to identify SVs and then compared the SV callsets to the ground truth to assess SV discovery accuracy for each SV caller. DeBreak achieved the highest F1 scores among the four tested SV callers in all five types of simulated SVs in PacBio datasets and similarly achieved the highest F1 scores in four SV types in Nanopore datasets (Table 1, Fig. S5a). Overall, DeBreak achieved an accuracy of 99.23% on simulated PacBio and 98.35% on simulated Nanopore data, which was higher than the other three SV callers. Simulated read length had a minor effect on DeBreak SV discovery accuracy, as DeBreak achieved a similar F1 score for all types of SVs in all three replicates (Table S1). All evaluated SV callers have a critical parameter, "minimum number of supporting reads", which determines the sensitivity of SV detection for these tools. In the PacBio simulation, we manually set the parameter of "minimum number of supporting reads" to a series of values for each caller and assessed the sensitivity of SV discovery at different thresholds. Although the recall of all SV callers dropped as the number of supporting reads increased, DeBreak consistently demonstrated the highest sensitivity at each threshold (Fig. S5b).

As SVs tend to emerge around repeats in the genome, we performed additional simulations of repeat-associated SVs. Four SV callers were applied to identify SVs and benchmarked to assess their abilities in resolving SVs located in repeats. DeBreak achieved an accuracy of 98.67% and 97.71% using PacBio and Nanopore data, respectively, which was higher than the other three SV callers (Table S2). In both insertion and deletion detection, DeBreak achieved the highest F1 score among four tested SV callers using both PacBio and Nanopore

data. The SV discovery accuracy for repeat-associated SVs was slightly lower than the previous random SVs for both insertions and deletions.

DeBreak implanted a large-insertion detection module to identify insertions longer than sequencing reads with local de novo assembly. To assess the improvement in maximal detectable insertion size of the large-insertion detection module, we embedded 1000 insertions with sizes ranging from 5 to 100 kbp into Chr1 and then simulated PacBio-like reads with a mean read length of 15 kbp. In general, insertion (INS) detection recall dropped as the insertion size increased for each SV caller, and DeBreak achieved the highest recall in each size category (Fig. S6). When the length of insertions exceeded sequencing reads (20k–30k), DeBreak identified 70% of homozygous INS and 40% of heterozygous INS, while the other three SV callers failed to detect any events. The maximal detectable insertion size of DeBreak was approximately twice the average read length, as the recall dropped dramatically when insertions were longer than 30 kbp.

We then investigated the accuracy of the detection of SV breakpoints. By refining SV breakpoints with the partial order alignment algorithm, DeBreak reconstructed the consensus sequences flanking the SVs, which showed much higher base accuracy than the raw reads (Fig. S7a). From the accurate consensus sequences, DeBreak can infer more precise SV breakpoints than merely from the raw reads (Fig. S7b). DeBreak identified 59.81% of SVs with exact breakpoint positions and 81.33% of SVs within 1 bp of the true SV breakpoint. We then downsampled the simulated datasets to assess the effect of sequencing depth on breakpoint accuracy. Overall, the breakpoint accuracy was improved when sequencing depth increased in both PacBio and Nanopore simulations, and DeBreak was able to achieve high breakpoint accuracy starting at 20x (Fig. S8). These results demonstrated that DeBreak can detect all five types of SVs in the simulated datasets with high accuracy.

### Benchmark on real human genome

We next benchmarked SV discovery accuracy on a real human genome, HG002, from the Genome in a Bottle (GIAB) Consortium[35]. We aligned PacBio CLR, HiFi, and Nanopore reads of HG002 to the human reference genome and applied four SV callers, DeBreak, Sniffles, pbsv, and cuteSV, on the three datasets. The GIAB community genome provided an SV callset of 4237 deletions and 5440 insertions from multiple platforms in defined "high-confidence" regions[36]. Thus, we first benchmarked the SV discovery accuracy in these high-confidence regions. In all three datasets, DeBreak achieved the highest SV

**Table 2 | SV discovery accuracy on HG002**

| | DeBreak | | | Sniffles | | | pbsv | | | cuteSV | | | PAV | | |
|---|---|---|---|---|---|---|---|---|---|---|---|---|---|---|---|
| | Rec | Pre | F1 | Rec | Pre | F1 | Rec | Pre | F1 | Rec | Pre | F1 | Rec | Pre | F1 |
| **Deletion** | | | | | | | | | | | | | | | |
| CLR | **97.73** | 96.48 | **97.10** | 95.14 | **96.69** | 95.91 | 95.28 | 96.21 | 95.74 | 97.31 | 94.18 | 95.72 | – | – | – |
| HiFi | **98.16** | 95.11 | **96.61** | 97.45 | 91.67 | 94.47 | 96.74 | 94.88 | 95.80 | 97.71 | 93.33 | 95.47 | 96.60 | **96.40** | 96.50 |
| Nano | **98.40** | **95.07** | **96.71** | 96.29 | 94.62 | 95.45 | 97.40 | 81.08 | 88.50 | 98.18 | 89.27 | 93.51 | – | – | – |
| **Insertion** | | | | | | | | | | | | | | | |
| CLR | **97.15** | **93.36** | **95.22** | 88.38 | 89.58 | 88.98 | 93.22 | 83.43 | 88.05 | 95.40 | 81.67 | 88.00 | – | – | – |
| HiFi | 97.26 | **92.84** | **95.00** | 90.90 | 87.79 | 89.32 | **97.41** | 80.42 | 88.10 | 96.64 | 89.88 | 93.14 | 96.18 | 91.32 | 93.69 |
| Nano | **97.46** | **93.91** | **95.65** | 90.57 | 90.01 | 90.29 | 95.07 | 85.60 | 90.09 | 96.99 | 89.27 | 92.97 | – | – | – |

The unit for recall, precision, and F1 score is %. The highest recall, precision, and F1 score among four tested SV callers are marked in bold.
*Rec* Recall, *Pre* Precision, F1 F1 score, *Nano* Nanopore, – not applicable.

discovery accuracy among the four tested SV callers, especially for insertions (Table 2). The higher SV discovery accuracy of DeBreak resulted from its advanced clustering algorithm, in which clustering window size is adjustable for SVs of different types, sizes, and local sequence content. Instead of setting a clustering window of fixed size, DeBreak computes the density of raw SV signals and determines the boundaries of the clustering window based on the density pattern. The clustering window is larger for longer SV events and smaller for shorter SV events, improving effectiveness by merging raw SV signals into SV candidates while excluding noisy signals nearby (Fig. S9). For SVs located in repetitive regions, the clustering window is automatically adjusted to tolerate shifts of raw SV signals caused by repeated segments (Fig. S10). We then stratified the SVs into different repeat classes in both ground-truth callset and SVs reported by four SV callers in HG002 to assess the SV discovery accuracy in each repeat type. Among the nine annotated repeat classes and non-repeat regions, DeBreak achieved the highest accuracy in 9 classes using CLR data, 5 classes using HiFi data, and 10 classes using Nanopore data, suggesting higher accuracy of DeBreak in resolving repeat-associated SVs (Fig. S11).

Previous work[37] has highlighted the functional importance of multi-allelic copy number variations (mCNVs) in gene dosage and gene expression. DeBreak can accurately identify multi-allele SVs (mSVs) in individual genomes. After density-based clustering, raw SV signals of candidate mSVs are further clustered through a k-means clustering algorithm to characterize two non-reference alleles (Fig. S12). We applied this method to identify putative mSVs in HG002. In total, we identified 802 multi-allele SVs in a single genome. The majority of mSVs (78/87, 89.66% in CLR; 49/53, 92.45% in HiFi; 74/90, 82.22% in Nanopore) in high-confidence regions had at least one allele matching with the ground-truth SV set. Multiple alternative alleles of the same SV event were probably merged into one allele in the high-confidence SV callset, while DeBreak can report both alleles (Fig. S13).

We also benchmarked genotyping accuracy of the four tested SV callers with the high-confidence SV callset. On the three datasets, DeBreak and cuteSV performed better than pbsv and Sniffles (Table S3). DeBreak achieved the highest genotyping accuracy in PacBio CLR and Nanopore datasets, while cuteSV achieved slightly higher genotyping accuracy in the PacBio HiFi datasets. We performed downsampling for PacBio CLR, HiFi and Nanopore datasets and assessed the genotyping accuracy in depth ranging from 10× to 100×. Greater sequencing depth of the input dataset increased DeBreak's genotyping accuracy, but data type had only a relatively minor impact on genotyping accuracy (Fig. S14).

We then assessed the SV discovery accuracy for SVs of different sizes. DeBreak achieved consistent and high accuracy for small and large SVs, especially for detecting insertions (Fig. 2a). Notably, for ultra-large insertions longer than the sequencing reads (>10 kbp), DeBreak achieved higher accuracy, recall, and precision than the other three SV callers (Fig. S15), benefiting from its large-insertion detection

module with local de novo assembly. In the PacBio HiFi and Nanopore datasets, DeBreak also achieved relatively high accuracy for SVs of different sizes (Fig. S16). We next evaluated the accuracy of SV breakpoint positions reported by DeBreak. The high sequencing error rate of long reads often causes imprecise inference of SV breakpoints. We compared SV callsets from the four tested SV callers to high-confidence benchmark SV callset to assess for shifts in breakpoint positions. With the breakpoint refinement module, DeBreak identified 59.90% of SVs with exact SV breakpoints and 63.53% of SVs with breakpoint shift within 1 bp as reported in GIAB, which was higher than pbsv (41.73% and 47.41%), Sniffles (4.99% and 13.45%), and cuteSV (5.18% and 13.87%) using the PacBio CLR dataset (Fig. 2b). For PacBio HiFi and Nanopore datasets, DeBreak also achieved the highest SV breakpoint accuracy among the four evaluated SV callers (Fig. S17).

To assess the effect of sequencing depth on SV discovery accuracy, we downsampled the PacBio CLR, HiFi, and Nanopore datasets by a series of depths by randomly sampling the reads. At each depth, DeBreak and pbsv were applied with the default parameters. In contrast, a set of parameters were tested for Sniffles and cuteSV, and the SV calls with the highest F1 scores were selected for comparison. Overall, SV discovery was more accurate for datasets having higher sequencing depth (Fig. 2c). Starting from 20×, DeBreak already achieved an accuracy of over 90% for PacBio CLR, HiFi, and Nanopore datasets (Fig. S18). For datasets with depth ≥20×, DeBreak consistently identified SVs with the highest accuracy among the four tested SV callers. Note that DeBreak and pbsv automatically adapted to lower depths using default settings, while Sniffles and cuteSV both required extra effort in manually tuning parameters to optimize performance. We further benchmarked the SV breakpoint accuracy in downsampled datasets. DeBreak reported most SVs with exact breakpoints starting at 20× in Pacbio HiFi and Nanopore datasets and at 30× for PacBio CLR data (Fig. S19). Taken together, these results highlight that DeBreak can accurately identify insertions and deletions, two major types of SVs, with precise breakpoints in real human genomes.

## Comparison to assembly-based SV discovery

Currently, de novo assembly is used to comprehensively characterize genome-wide SVs[27–30]. To compare alignment-based with assembly-based SV discovery approaches, we applied DeBreak, pbsv, Sniffles, and cuteSV on six samples from the Human Genome Structural Variation Consortium (HGSVC). Three of these six samples were sequenced with the PacBio CLR platform, and the other three were sequenced with the PacBio HiFi platform. For these samples, highly accurate assembly-based SV callsets were generated by performing haplotype-resolved de novo assembly with phased sequencing reads and subsequent SV discovery from whole-genome assembly with the PAV pipeline[27]. Overall, the assembly-based SV approach discovered a slightly higher number of SV events (22,897–27,187) compared with alignment-based methods. By treating these SVs as the "ground truth",

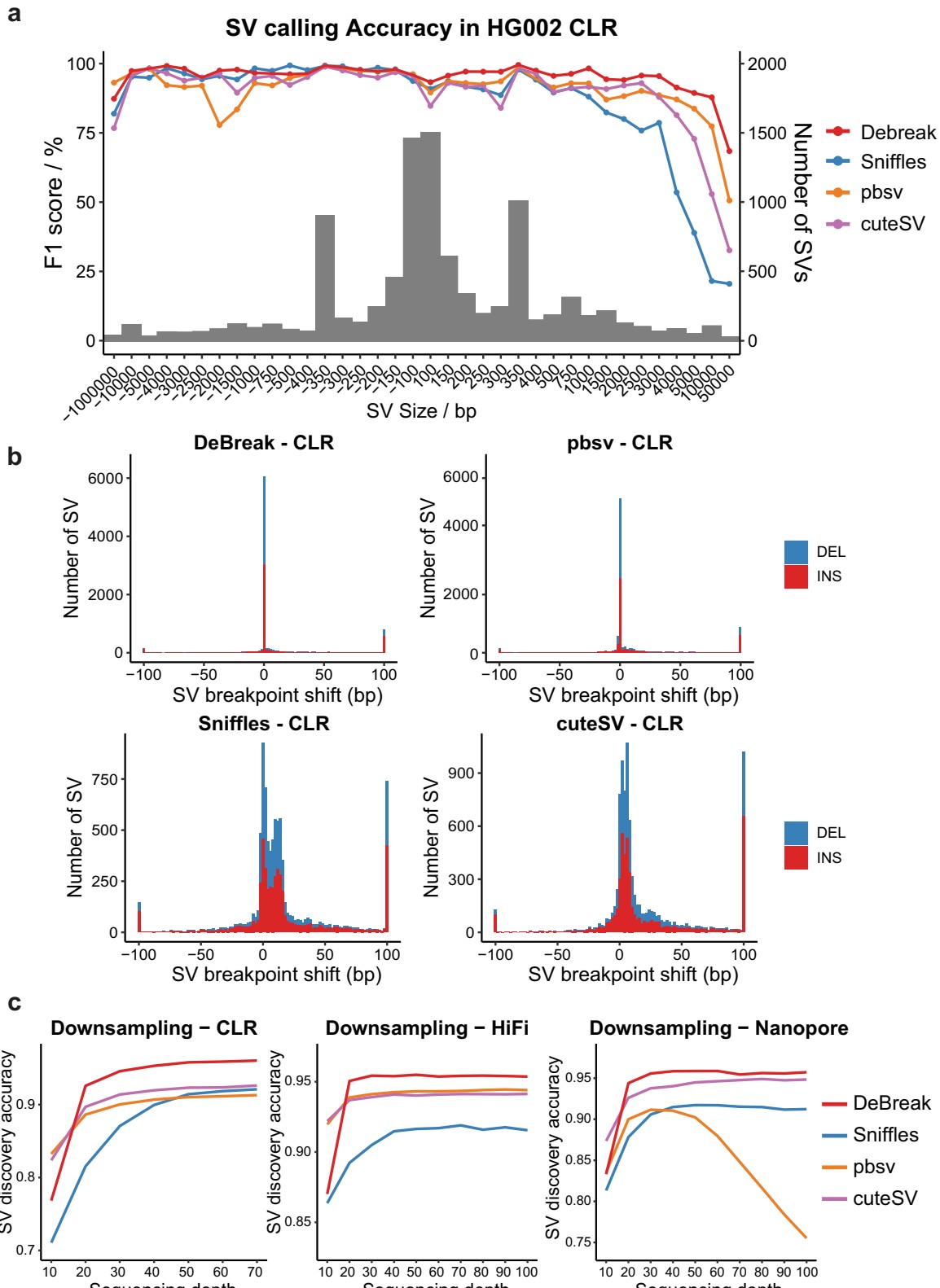

**Fig. 2 | SV discovery in HG002. a** SV discovery accuracy for insertions (positive SV size) and deletions (negative SV size) at different size ranges in the CLR dataset. Bars indicate the number of SVs in each size range, and lines show the SV discovery accuracy for each SV caller. **b** SV breakpoint accuracy for four tested SV callers in the CLR dataset. SVs with breakpoint shifting >100 bp were included in the ±100 bp bins. **c** SV discovery accuracy in downsampled PacBio CLR (left), HiFi (middle), and Nanopore (right) datasets. Source data are provided as a Source Data file.

we evaluated the SV discovery accuracy of four alignment-based SV callers. DeBreak identified SV with an average F1 score of 80.09% in the six samples, which was higher than pbsv (72.68%), Sniffles (71.44%), and cuteSV (77.38%) (Tables S4 and S5). In each sample, DeBreak achieved both higher recall and precision than the other three callers (Fig. 3a), suggesting a higher consistency with the assembly-based SV discovery than the other three alignment-based SV callers. Among four tested SV callers, cuteSV showed the highest consistency in SV genotyping with an assembly-based approach (Table S6). DeBreak detected a total of 3100 mSVs in the three CLR datasets and 3097 mSVs in the three HiFi datasets, with ~71% of these 6200 and 6194 alternative alleles validated by assembly-based approaches in CLR and HiFi datasets, respectively (Table S7). A total of 23 and 24 mCNVs were identified in the three CLR and HiFi datasets with further annotation of copy number variation using $k$-mer counts (Fig. S20).

As assembly-based SV calls usually have accurate SV breakpoints inferred from assembled contigs, we also compared the SV breakpoint accuracy of four SV callers. With the breakpoint-refinement module, DeBreak identified 46.83% of SVs with exact SV breakpoints and 48.12% of SVs within 1 bp shift on the three PacBio CLR datasets, while pbsv reported 39.03% and 40.99%, Sniffles reported 3.02% and 6.82%, and cuteSV reported 2.75% and 7.69% of SVs with exact breakpoints and within 1 bp shift, respectively (Fig. 3b). For the three PacBio HiFi datasets, DeBreak also achieved better breakpoint accuracy than the other three callers, as 56.98%, 47.41%, 2.99%, and 15.33% of SVs were identified with exact SV breakpoints, and 58.09%, 48.85%, 6.41%, and 35.54% of SVs were identified within 1 bp shift by DeBreak, pbsv, Sniffles, and cuteSV, respectively.

Approximately 82% of DeBreak SV calls overlapped with the assembly-based SV callset. There are several thousand unique SVs were reported either by DeBreak or by the assembly approach. To characterize these SVs, we performed a four-way comparison of SV callsets from DeBreak, pbsv, cuteSV, and PAV on the sample HG00096. For the SVs identified by PAV but not by DeBreak, 27.1% of deletions and 30.5% of insertions were reported by either pbsv or cuteSV (Figs. 3c and S21). In contrast, 71.9% of deletions and 71.5% of insertions reported by DeBreak but not PAV were also reported by either pbsv or cuteSV. We further characterized the 3385 SVs only reported by PAV and 1292 SVs only reported by DeBreak in all six samples. Note that DeBreak reported the fewest number of unique SVs. By examining the SV locations on the genome, we found that there was strong enrichment in telomere regions for PAV-unique SVs (43.5% located near telomeres, 5.8% located near centromeres, and 46.4% located in repetitive regions) (Figs. 3d and S22). While DeBreak-unique SVs were enriched in the telomere and centromere regions (31.2% located near telomeres, 27.7% located near centromeres, and 38.7% located in repetitive regions). Although DeBreak controls read depth and a minimum number of supporting reads, alignment-based SV discovery may have inaccurate read alignment in these regions. However, it is also challenging to assemble reads with abnormal coverage and ascertain the phasing status of individual reads without bias. Additional efforts are needed to validate these SVs.

We then compared four alignment-based SV callers to an assembly-based approach in the CHM13 cell line, where a complete telomere-to-telomere assembly is available[38]. An assembly-based SV callset was generated using Dipcall[39] on the CHM13 assembly. We selected the high-confidence regions of Dipcall as "ground truth". Four alignment-based SV callers were applied on the PacBio CLR (70×), HiFi (57×), and Nanopore (126×) read alignment files. For both insertion and deletion detection, DeBreak achieved the highest consistency with assembly-based SV callset in all three data types (Table S8). All SVs in CHM13 should be homozygous, as CHM13 is a haploid cell line. We then benchmarked the genotyping accuracy of four alignment-based SV callers, with only 'GT = 1/1' as the correct genotype. In CHM13, DeBreak achieved the highest genotyping accuracy in PacBio HiFi and

Nanopore datasets, and cuteSV achieved slightly higher genotyping accuracy in PacBio CLR dataset (Table S9).

## SV discovery in cancer genomes

SVs play important roles in cancer development and progression[40–42]. Unlike germline SVs, cancer genomes may contain more large-scale deletions, duplications, inversions, translocations, and other complex SVs[43–45]. DeBreak includes a "tumor" mode to identify "abnormal" SVs and SVs with clustered breakpoints in cancer genomes. To assess SV discovery in cancer genomes, we applied the four SV callers to identify SVs in a breast cancer cell line, SKBR3. The PacBio CLR dataset (72×, mean read length of 9.87 kbp) of SKBR3 was downloaded and aligned to the human reference genome. Under the "tumor" mode, DeBreak identified 8249 deletions, 9226 insertions, 3129 duplications, 190 inversions, and 137 translocations. We compared the SV callsets from the four SV callers. As expected, large proportions of SVs were identified by all four SV callers (Fig. S23). Among the four SV callers, DeBreak reported relatively fewer singleton SV calls, especially in insertion/duplication detection, suggesting the high precision of DeBreak SV callsets. We also compared the SV callset of DeBreak to previously reported SV lists from long-read and short-read data[46] (Fig. S24). DeBreak reported additional 1333 deletions, 3073 insertion/duplications, 51 inversions, and 91 translocations. To validate potential cancer-related SVs reported only by DeBreak, we designed primers flanking breakpoints for 15 randomly selected SVs (6 deletions, 4 duplications, 3 inversions, and 2 translocations) that spanned more than 10 kbp (Fig. S25). Polymerase chain reaction (PCR) experiments validated 12 out of 15 DeBreak-unique SVs, with a validation rate of 80% (Supplementary Data 1).

We further analyzed SVs in the SKBR3 breast cancer cell line by annotating breakpoints and identified 41 putative gene fusions. By cross-validating these gene fusion events with Iso-Seq data (see the "Methods" section), we found 11 gene fusions that can be validated at the transcripts level (Supplementary Data 2). The cross-validation rate was 26.83%, which was higher than Sniffles (25/116, 21.55%), pbsv (5/39, 12.82%), and cuteSV (7/58, 12.07%). 6 out of the 11 cross-validated gene fusions identified by DeBreak have been previously reported using transcriptomic data[46–49]. Therefore, SV discovery using DNA-seq data with DeBreak identified five additional gene fusions: WDR82-PBRM1, PDE4D-DEPDC1B, CPNE1-PHF20, CSE1L-KCNB1, and CSNK2A1-NCOA3. The fusion of WDR82 and PBRM1 was caused by a hemizygous deletion of 392 kbp on chromosome 3, with the fusion junction located in the intronic region of both genes (Fig. S26). A deletion of 259 kbp on chromosome 20 caused the fusion of CSE1L and KCNB1, where the seventh exon of CSE1L was fused with the intron of KCNB1 (Fig. S27). The gene fusion junction locations observed in the Iso-Seq reads were highly consistent with SV breakpoint positions inferred by DeBreak, suggesting that DeBreak can accurately predict SV breakpoint positions in cancer genomes. These results indicate that DeBreak can be applied to cancer genomes and comprehensively identify different types of SVs.

## Runtime and memory usage

DeBreak and other SV callers were tested on Intel Xeon E5-2680 v3 CPUs with 12 cores and 2.5 GHz of frequency. It took 12.4 h for DeBreak to identify SVs from a human genome (SKBR3 cell line) using the 67x PacBio CLR dataset with a peak memory of 63 GB (Table S10). Due to the local assembly module and partial order alignments, DeBreak consumed more runtime and memory than Sniffles (3.0 h, 13GB) and cuteSV (1.5 h, 3 GB). However, DeBreak was much faster and consumed less memory than pbsv (45.1 h, 72 GB).

## Discussion

In this work, we present DeBreak, a method for efficient and accurate structural variation detection from long-read sequencing data. Based

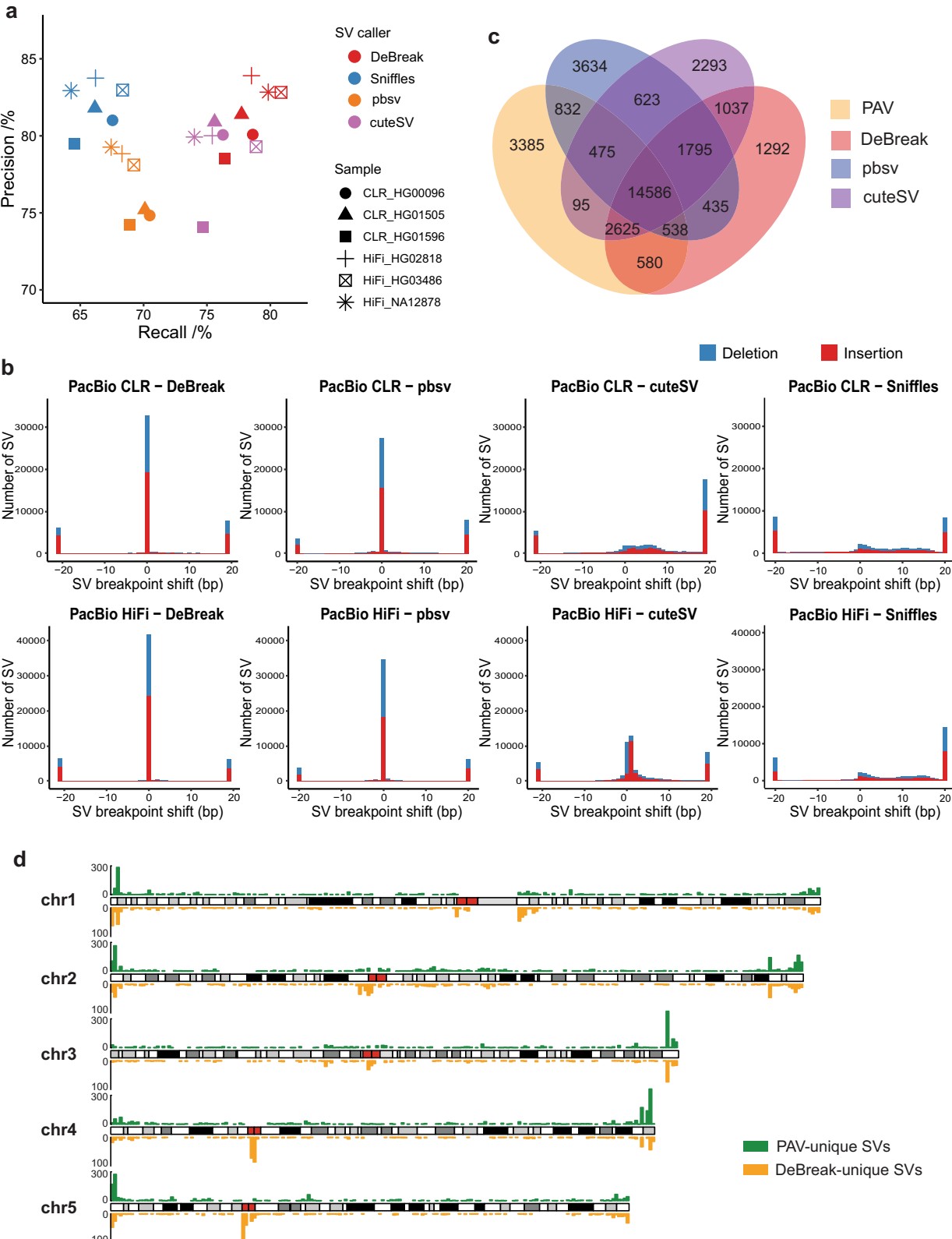

**Fig. 3 | Alignment-based and assembly-based SV discovery. a** SV discovery recall and precision of alignment-based SV callers when compared with the assembly-based SV callset. **b** SV breakpoint accuracy of DeBreak, pbsv, cuteSV, and Sniffles in PacBio CLR (top) and HiFi (bottom) datasets. **c** Venn diagram showing the overlap among four SV callsets. The number of SV events in each category is labeled within each section. **d** Distribution of PAV-unique and DeBreak-unique SV calls on chromosomes 1–5. Red boxes indicate the positions of centromeres. Source data are provided as a Source Data file.

on simulation data, real human genome data, and cancer cell line data, DeBreak has demonstrated excellent performance when compared with several state-of-the-art long-read SV callers. The improved performance is due to several innovative design features: (1) the density-based clustering method can accurately identify candidate SV events with a variety of sizes; (2) the partial order alignments can produce a consensus sequence for accurate breakpoint inference, which is helpful for experimental validation and mechanism inference[21–24]; (3) local de novo assembly facilitates the discovery of long insertion events, which usually cannot be inferred within individual reads; (4) k-means approach can accurately identify multi-allele SVs, which are functionally important; and (5) multiple functions can be applied to both healthy and unhealthy genomes.

Due to the limited availability of ground-truth SV sets, DeBreak was benchmarked for insertion and deletion discovery in HG002 and HGSVC samples, but not for duplication, inversion, or translocation. Further validation of SV discovery accuracy on these SV types would be desirable and will help improve DeBreak's performance if comprehensive high-confidence truth SV sets become more readily available. Although the benchmark was based on human genomes, DeBreak can be applied to other diploid or haploid non-human long-read resequencing data. The overall workflow may be applied to polyploid genomes as well. Based on our knowledge and experience, SV discovery in polyploid genomes is challenging for any currently available tools. More sophisticated benchmarking work is needed.

Several features of the input sequencing dataset have an essential impact on SV discovery accuracy. Data type (sequencing platform) affects SV discovery accuracy and breakpoint accuracy. Based on our benchmarks and as expected, datasets with lower sequencing error rates often lead to better SV discovery accuracy and breakpoint accuracy than datasets with higher error rates at similar levels of sequencing depth. Sequencing depth also affects accuracy for SV discovery, breakpoint position, and genotyping. Sequencing read length can affect the maximal size of detectable SVs, especially for insertion detection.

We observed that SV callers reporting more accurate breakpoint positions (DeBreak and pbsv) required more computational resources than SV callers with less accurate breakpoints (Sniffles and cuteSV). During SV discovery for DeBreak, breakpoint refinement and ultra-large insertion detection were the two most time-consuming steps, accounting for approximately 45% and 32% of total runtime, respectively. When we disabled these two features, DeBreak accomplished SV detection within 2.8 h for the same sample, similar to the runtime of Sniffles and cuteSV. The extra runtime and memory usage helped improve the quality and accuracy of the DeBreak SV callset. In all situations, DeBreak and other alignment-based methods consume much less computational resources than assembly-based methods. Although comprehensive evaluation and validation between alignment-based and assembly-based approaches are needed, alignment-based methods will continue to serve important roles in SV analysis.

## Methods

### DeBreak workflow

**Overall workflow of DeBreak.** DeBreak detects SVs from read-to-reference alignments generated by any long-read aligner, such as minimap2, pbmm2, and ngmlr. The workflow of DeBreak includes (1) raw SV signal detection, (2) large insertion identification, (3) SV signal clustering, (4) multi-allele SV identification, (5) SV breakpoint refinement, and (6) SV filtering and genotyping. The output of DeBreak is a standard VCF file containing confident SV calls.

**Raw SV signal detection and clustering.** Raw SV signals are detected from read-to-contig alignment. DeBreak scans all read alignments for intra-alignment and inter-alignment SV signals. Smaller insertions

and deletions can be contained within a single alignment (Fig. S1a). For larger indels, inversions, duplications, and translocations, DeBreak utilizes split-read information and classifies SV type based on orientation and clipping location of two segments from the same read (Fig. S1b). Insertions are inferred when there are extra sequences in the read between two adjunct alignments. Deletions are inferred when a region on the reference genome is skipped between two alignments. Duplications are inferred when two alignments are overlapped on the reference genome. Inversions are inferred when two alignments have distinct orientations. Translocations are inferred when the read is aligned to two distinct chromosomes with help of "SA" tag in the BAM file. As it scans through read alignments, DeBreak also estimates the sequencing depth of the input dataset and automatically adjusts parameters used in the following clustering and filtering processes.

Raw SV signals are then clustered into SV candidates using a density-based clustering algorithm for insertion, deletions, duplications, and inversions (Fig. S2). All signals from the same chromosome with the same SV type are sorted based on coordinates. The density of SV raw signals is computed for each position on the chromosome. DeBreak scans the chromosome for density peaks above the threshold, which is automatically adjusted according to the sequencing depth of the input dataset. For each peak, the boundaries of the SV region are defined on both sides of the peak summit when the density drops to 10% of the summit height. All raw signals located within the SV region are then merged into one SV candidate. For translocation, positions of both breakpoints are clustered with a fixed window of 400/800 bp. The window size is determined by the standard variation of breakpoint positions. A 400 bp window is used for groups of raw signals with smaller standard variation and an 800 bp window is used for groups of raw signals with a larger standard variation.

For each SV candidate, DeBreak determines whether it is a multi-allele SV based on the first quartile (Q1) and third quartile (Q3) of SV size from all raw signals. If Q3 is smaller than twice of Q1, all raw signals are merged into a single-allele SV, excluding outliers of extremely large or small size. If Q3 is larger than twice Q1, DeBreak separates raw SV signals for each allele with k-means clustering (k = 2 for diploid genomes) and merges signals from each cluster separately as a multi-allele SV candidate. The detection and clustering of SV signals are processed separately for each chromosome, allowing DeBreak to perform multi-thread SV detection, drastically reducing runtime.

**SV breakpoint refinement.** After SV signal clustering, DeBreak assigns each SV candidate a breakpoint coordinate by computing the mean value of raw signals. Raw signals can be highly imprecise due to the high error rate of long-read sequencing and the presence of low-complexity regions in the genome. DeBreak implants the POA algorithm from wtdbg2[50] to refine breakpoint locations. For each SV candidate, DeBreak collects all reads containing raw signals of this SV candidate and performs POA to generate accurate consensus sequences. DeBreak then aligns these consensus sequences to the reference genome with minimap2 and detects SVs from consensus sequence alignments. The breakpoint location detected from the consensus sequence is used to refine the breakpoint coordinates of SV candidates. If POA fails to generate consensus sequences for an SV candidate, or the consensus sequence cannot be properly aligned back to the genome, DeBreak will keep the mean value of the raw signals as breakpoint coordinates.

**Depth-based filtering and genotyping.** During raw SV signal detection, DeBreak records the total length of aligned reads on each chromosome and computes the average sequencing depth. Reads containing raw signals of a particular SV event are considered 'supporting reads' for this SV. The minimum threshold of supporting reads ($N_{supp}$) is determined based on the average sequencing depth:

$N_{\text{supp}} = \frac{\text{Depth}}{10} + 2$. SV candidates supported by at least $N_{\text{supp}}$ reads are kept for further consideration, and the rest are discarded to remove background noise. SVs of low mapping quality is also filtered to remove false positives caused by inaccurate read alignment. For multi-allele SVs, DeBreak filters each allele independently. If only one allele passes, a single-allele SV will be reported instead. SVs are genotyped based on the ratio of SV supporting reads to the local sequencing depth at each SV location.

**Large insertion detection via local assembly.** DeBreak utilizes a local de novo assembly approach to detect ultra-large insertions that are too long to be spanned within single reads. While scanning read alignments for raw SV signals, DeBreak also records the positions of clipped ends of read alignments. Read alignments with at least 200 bp unmapped sequences (clipped sequences) on either side are considered 'clipped' alignments (Fig. S3a). After scanning through a chromosome, DeBreak identifies candidate insertion breakpoint regions with enriched clipped alignment, where at least $N_{\text{supp}}$ reads are clipped on the left side of the candidate breakpoint and another $N_{\text{supp}}$ reads are clipped on the right side of the breakpoint. It then collects these clipped reads at each candidate breakpoint region and performs local de novo assembly with wtdbg2 to reconstruct assembly contigs that contain full-length inserted sequences (Fig. S3b). DeBreak aligns assembled contigs to the reference genome with minimap2 and detects insertions from these contigs. Detected insertions are filtered out if (1) multiple contigs are assembled during local de novo assembly, (2) a detected insertion is located in another chromosome or too far away from the candidate insertion breakpoint, or 3) the detected insertion is smaller than 1kbp.

**Duplication identification.** DeBreak includes an optional duplication-rescuing module that distinguishes tandem duplications from insertion calls, as smaller tandem duplications are often treated as insertions by aligners. The inserted sequence of tandem duplication shows high similarity with the duplicated region, while insertions are usually consistent with novel sequences or sequences from distinct regions of the genome. For each insertion call, DeBreak collects reads supporting the SV event and extracts inserted sequences from each read. It utilizes minimap2 to re-align these inserted sequences back to the local region (1 kbp flanking the insertion breakpoint) on the reference genome. If more than 50% of inserted sequences can be aligned back to the local region, DeBreak corrects the SV type to tandem duplication for this insertion call.

**Benchmark in simulated dataset**

**Simulated dataset generation.** Three simulated datasets with ground-truth SVs were generated for benchmarking. For each dataset, a total of 22,200 SVs (10,000 deletions, 10,000 insertions, 1000 duplications, 1000 inversions, and 200 translocations) were randomly simulated on Chr1 to Chr22 and ChrX. The sizes of simulated SVs followed the geometric distribution as observed in real human genomes, including peaks at ~350 and ~6000 bp. These simulated SVs were assigned as heterozygotes and homozygotes with a ratio of 2:1, and heterozygous SVs were randomly assigned to two haplotypes. The human reference genome GRCh38 (autosomes and the X chromosome) was modified according to the type and size of simulated SVs to generate haplotype 1 and haplotype 2. PacBio-like reads were simulated from the modified genome using pbsim (v1.0.3) with options "–data-type CLR –model_qc model_qc_clr –depth 25 –accuracy-mean 0.85". Nanopore-like reads were simulated using Badread (v0.2.0) with options "–quantity 25x –junk_reads 0 –random_reads 0 –chimeras 0 –glitches 0,0,0". The depth was set to 25X for each haplotype, generating a simulated dataset of 50X when merging all reads from both haplotypes. The average read length was set to 10, 15, and 20 kbp for three simulated datasets.

**SV discovery in simulated datasets.** The simulated reads were aligned to the reference genome with minimap2 (v2.15), ngmlr (v0.2.7), and pbmm2 (v1.3.0) under default settings. DeBreak (v1.2) was applied to minimap2 alignment results with default settings. pbsv (v2.6.2) was run on pbmm2 alignment results with default settings, and Sniffles (v1.0.8) was run on the ngmlr alignment results with options "–genotype -s 4/5/6/7/8/9/10". A serious of -s (minimum number of reads supporting an SV) was tested for Sniffles, and the threshold with best accuracy was selected for comparison with other SV callers. cuteSV (v1.0.11) was run on minimap2 alignment results with options "–genotype". All SVs with lengths≥45 bp were selected for benchmark.

The SV callsets of DeBreak, Sniffles, pbsv, and cuteSV were compared to the ground-truth SV set to assess the recall, precision, and F1 score. An SV call (DEL, INS, DUP, and INV) is considered a true positive (TP) if all three conditions are met:

1. $\text{Type}_G = \text{Type}_C$
2. $\text{ABS}(\text{Cor}_G - \text{Cor}_C) \leq 1\,\text{kbp}$
3. $0.5 * \text{Size}_G \leq \text{Size}_C \leq 2*\text{Size}_G$

Where the $\text{Type}_G$, $\text{Type}_C$, $\text{Cor}_G$, $\text{Cor}_C$, $\text{Size}_G$, and $\text{Size}_C$ are the SV type, start coordinates, and size of the ground truth SV call and the candidate SV call. For translocations (TRAs), the coordinates of both breakpoints on two chromosomes should be within 1 kbp flanking the ground-truth breakpoints to be determined as TP.

**Simulation of repeat-associated SVs.** RepeatMasker annotation of the human genome was downloaded from UCSC Table Browser. 10,000 repeats were randomly selected with sizes ranging from 50 bp to 20 kbp. Insertions were simulated by adding an additional copy of the repeat, and deletions were simulated by removing the repeat. SVs were assigned as 'homozygous and 'heterozygous' with a ratio of 1:1. PacBio-like and Nanopore-like reads were simulated using pbsim (v1.0.3) and Badread (v0.2.0) with a sequencing depth of 50x and average read length of 10, 15, and 20 kbp. Sequencing reads were aligned to the human reference genome with minimap2 (v2.15), ngmlr (v0.2.7), and pbmm2 (v1.3.0). DeBreak (v1.2) and pbsv (v2.6.2) were applied on read alignment files with default settings. A series of "-s" were provided for Sniffles (v1.0.8) and cuteSV (v1.0.11), and the SV callsets with the highest accuracy were used for comparison. SV discovery accuracy was benchmarked using the same criteria as in the section "SV discovery in simulated datasets".

**Simulation of ultra-large insertion.** 1000 insertions were randomly simulated and embedded into human Chromosome 1, with insertion sizes ranging from 5 to 100 kbp. Insertions were assigned as 'homozygous and 'heterozygous' with a ratio of 1:2. 50x PacBio-like reads were simulated with an average read length of 15 kbp and then aligned to the human reference genome. DeBreak (v1.2), Sniffles (v1.0.8), pbsv (v2.6.2), and cuteSV (v1.0.11) were applied on read alignment files to identify SVs with default settings. Recall for insertion detection was benchmarked at different size ranges using the same criteria as in the section "SV discovery in simulated datasets".

**Benchmark in HG002 dataset**

**SV discovery accuracy benchmark.** Raw sequencing reads (PacBio CLR, HiFi, and Nanopore data) were downloaded and aligned to GRCh37 with minimap2 (v2.15)[51], ngmlr (v0.2.7), and pbmm2 (v1.3.0) with default settings. DeBreak (v1.2) and cuteSV (v1.0.11) were applied to minimap2 alignment with default settings. pbsv (v2.6.2) was applied to pbmm2 alignment with default settings. Sniffles (v1.0.8) was applied to ngmlr alignment with the option "–genotype -s 9/9/12" for PacBio CLR, HiFi, and Nanopore datasets, respectively. A series of minimal supporting read (-s option) were tested for Sniffles, and the callset with the best performance was used for evaluation. Coordinates of SVs in the PAV callset were converted from hg38 to hg19 using LiftOver. SV

callsets of four alignment-based SV callers and PAV were benchmarked within the high-confidence regions (HG002_SVs_Tier1_v0.6.bed) by comparing to the benchmark SV callset using the same criteria as for the simulation benchmark. Repeat types of SVs were classified with RepeatMasker (v4.1.2) using sequences of the longest allele. Shifts of SV breakpoints were also evaluated with the high-confidence SV benchmark callset. The SV coordinates in DeBreak and cuteSV callsets are 1-based, so all the breakpoint positions were transformed to 0-based to keep consistent with the benchmark callset. Genotyping accuracy of four SV callers was evaluated based on the genotype information in the benchmark callset.

**Down-sampling.** To evaluate SV callers at varying sequencing depths in HG002, we downsampled the PacBio CLR dataset to a series of depth from 10x to 70x and downsampled PacBio HiFi and Nanopore datasets to a series of depths from 10x to 100x. Sequencing reads were randomly selected to generate datasets of the desired depth. The depth of each down-sampled dataset was validated by the total number of bases in reads divided by human genome size (3.1 Gbp). Four SV callers were first applied to downsampled datasets with default settings. In addition, to achieve the best performance of Sniffles and cuteSV, a series of min_supp (-s option) was provided to Sniffles and cuteSV at each depth, and the SV callset with the highest accuracy was selected for comparison.

**Comparison with assembly-based SV callset**
**SV discovery in HGSVC samples.** Raw PacBio CLR or HiFi reads of sample HG00096, HG01505, HG01596, HG02818, HG03486, and NA12878 were downloaded and aligned to GRCh38 with minimap2 (v2.15), ngmlr (v0.2.7), and pbmm2 (v1.3.0) under default settings. DeBreak (v1.2), pbsv (v2.6.2), Sniffles (v1.0.8), and cuteSV (v1.0.11) were applied to the alignment files to identify SVs with default settings. The merged assembly-based SV callset was downloaded from HGSVC2 data portal, and SVs of each sample was extracted with custom script. The comparison of SV calls was performed for autosomes and the X chromosome. SVs located within 5 Mbp of both ends of the chromosomes were classified as 'near telomere'. SVs located within 5 Mbp of the centromere were classified as 'near centromere'. Remaining SVs were annotated according to the repeat annotation from Table Browser. SV Distribution on the genome was plotted with karyoploteR[52].

**SV benchmark in CHM13 cell line.** Dipcall (v0.3) was applied on the Telomere-to-Telomere assembly of CHM13 with default settings to generate an assembly-based callset. SVs with a size of at least 50 bp were used as ground truth callset. PacBio CLR, HiFi, and Nanopore reads were downloaded and aligned to GCh38 with minimap2 (v2.15), ngmlr (v0.2.7), and pbmm2 (v1.3.0). DeBreak (v1.2), Sniffles (v1.0.8), pbsv (v2.6.2), and cuteSV (v1.0.11) were applied on read alignment files with default settings. SV callsets were benchmarked only in the high-confidence regions suggested by Dipcall. Multimatch was allowed when comparing alignment-based SV callsets to the ground truth. Genotyping accuracy was benchmarked with 'GT = 1/1' as correct and the remaining as incorrect genotypes.

**SV validation in SKBR3 cell line**
**PCR validation of SVs identified by DeBreak only.** PCR validation was performed for SVs identified by DeBreak that were not reported previously by Sniffles and short-read SV callers[46]. Fifteen putative cancer-related SVs were randomly selected from SVs spanning more than 10 kbp on the genome. Insertions were not validated due to the length limitations of PCR. PCR primers were designed for each type of SV with Primer3 (v0.4.0)[53], and the specificity was verified with UCSC in-silico PCR (Fig. S25). An SV event was validated if the PCR and following gel electrophoresis confirmed PCR product of the predicted size.

**Gene fusion annotation and validation with Iso-Seq data.** PacBio CLR sequencing data of SKBR3 was aligned to GRCh38 for SV discovery with DeBreak. Breakpoints of deletions, duplications, inversions, and translocations were annotated based on the Ensembl GRCh38 annotation (v104). An SV was considered to cause gene fusion when its two breakpoints were located within two different genes. Iso-Seq reads were downloaded from NCBI and aligned to GRCh38. For each gene fusion event, the total number of Iso-Seq reads aligned to both genes was counted. Gene fusion events supported by at least three Iso-Seq reads were considered validated.

## Data availability

PacBio CLR, HiFi, and Nanopore HG002 sequences are publicly available at GIAB (https://github.com/genome-in-a-bottle/giab_data_indexes), where PacBio 70x (CLR), PacBio CCS 15kb_20kb chemistry2 (HiFi), and Oxford Nanopore ultralong were used for SV discovery. The Tier1 benchmark SV callset and high-confidence HG002 region were obtained from https://ftp-trace.ncbi.nlm.nih.gov/ReferenceSamples/giab/data/AshkenazimTrio/analysis/NIST_SVs_Integration_v0.6/. Sequencing reads and assembly-based SV callsets of HG00096, HG01505, HG01596, HG02818, HG03486, and NA12878 are publicly available on the HGSVC2 data portal at https://www.internationalgenome.org/data-portal/data-collection/hgsvc2. T2T assembly and sequencing reads of CHM13 are publicly available at https://github.com/marbl/CHM13. The PacBio CLR and Iso-Seq data of the SKBR3 cell line are publicly available at NCBI SRA under BioProject PRJNA476239. SV callsets evaluated in the paper are available at https://zenodo.org/record/7214225. Source data are provided with this paper.

## Code availability

DeBreak is publicly available at https://github.com/Maggi-Chen/DeBreak under the MIT License[54]. We used v1.2 version for SV discovery and benchmark presented in the manuscript. Key custom Python scripts used in the manuscript are available at https://github.com/Maggi-Chen/DB_code.

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

## Acknowledgements

This work was supported by a grant from the National Institute of General Medical Sciences (1R35GM138212); the BioData Catalyst Fellowship from National Heart, Lung, and Blood Institute (a sub-award from 1OT3HL147154) to Z.C.; and the Center for Clinical and Translational Science grant from the National Center for Advancing Translational Sciences (UL1TR003096) to A.Y.W.

## Author contributions

Z.C. conceived and managed the project. Y.C. implemented the algorithm, collected all datasets, and performed primary data analysis. Z.C., Y.Z., and M.G. were involved in data analysis and testing of the algorithm, and Z.C., A.Y.W., and M.G. were involved in the interpretation of the results. Y.C., Z.C., and A.Y.W. wrote the manuscript. X.Z., C.A.B., and M.D.E. performed experimental validation. All authors have read and approved the final manuscript.

## Competing interests

The authors declare no competing interests.
