## [Transparent Peer Review File · Nature Communications]

Deciphering the exact breakpoints of structural variations using long sequencing reads with DeBreakEDITORIAL NOTE

The responses on this document used screenshots from the Integrative Genomics Viewer (IGV). References:

Robinson, J., Thorvaldsdóttir, H., Winckler, W. *et al.* Integrative genomics viewer. *Nat Biotechnol* **29**, 24–26 (2011).

Robinson, J., Thorvaldsdóttir, H., Wenger, A. *et al.* Variant Review with the Integrative Genomics Viewer (IGV). *Cancer Research* **77**(21) 31-34 (2017).

REVIEWER COMMENTS

Reviewer #1 (Remarks to the Author: Overall significance):

Chen et al described DeBreak, a new structural variation (SV) caller for long sequence reads. DeBreak differs from the existing SV callers in its use of local reassembly, which I think is the right direction. The authors show that DeBreak outperforms other popular SV callers on both simulated and real datasets.

Reviewer #1 (Remarks to the Author: Impact):

In my opinion, this manuscript could be a fit to Nature Communications.

Reviewer #1 (Remarks to the Author: Strength of the claims):

Major comments:

1) Please make DeBreak support python3. Python2 has retired for more than a year. Users would question the long-term commitment to DeBreak if they see it support python2 only. Python2-only also makes it difficult for others to contribute to DeBreak as there will be fewer Python2 programmers in future.

2) It would be good to stratify the result by repeat classes. For example, what is the accuracy for ALUs, LINE1s, SVAs, STRs, VNTRs and non-repeats? I predict that the accuracy of every caller will be near perfect for ALUs and LINE1s and will drop a lot in VNTRs. It is rare to see such stratification in SV caller papers, but I think this is an important analysis and is likely to benefit this manuscript. The authors can run RepeatMasker/TRF on the longest allele to annotate repeats.

3) I am a little concerned with the low consistency between DeBreak and PAV. Could the authors compare PAV HG002 calls to GIAB? It would be important to understand the accuracy of PAV. Another option is to use dipcall for assembly-based SV calling. Dipcall is known to agree with GIAB well. Its accuracy is lower than read-based SV calls mostly due to different variant representations. Note that dipcall also generates confident regions like GIAB.

Minor comments:

4) What assembler is used for local assembly? Is it wtdbg2?

5) Does DeBreak assemble all reads mapped to a candidate SV, or only assemble reads that contains the SV?

6) What is the tool and the command line for comparing SV callsets? Is it truvari?

7) The last two pages on both "Supplementary file"s are not properly formatted. I guess this is generated by PDF printing. It would be good to have Excel files instead as it is difficult to derive a text file from PDF.

Reviewer #1 (Remarks to the Author: Reproducibility):

I could install and run DeBreak via Bioconda but I have not tried it on large-scale datasets.

Reviewer #2 (Remarks to the Author: Overall significance):

Firstly my apologies to the authors for a slow review.

I have attempted to run DeBreak on some samples and have found it to be a useful tool. I hope that this tool will be used in future. However if the authors wish it to be used widely it will require some tuning. We were able to identify known events in samples using DeBreak that were not identified in either cuteSV or sniffles (VERSION 2). This is useful and I have already added DeBreak to my suite of tools to use to look for SVs.

Reviewer #2 (Remarks to the Author: Impact):

As I state below, I am already investigating using DeBreak to look at SVs alongside tools such as CuteSV and Sniffles. The analysis of SVs detected using long read technology is of huge interest and the tools available to do so are still maturing. DeBreak is a worthy tool in this suite of methods.

Reviewer #2 (Remarks to the Author: Strength of the claims):

1) Installation 1: I do have several comments about the code and its implementation that have been challenging. Firstly, the code is written in python 2.7 - now end of life (not supported). Really the tool should be updated to a current version of python. Similarly, the dependencies required for installing are old with all having been updated in the last two years versus the tested versions described.

2) Installation 2: We also could not get the code to install using the suggested conda instructions - instead we had to install in a specific environment file (see below for the yaml file we created to make an environment). This may have been a peculiarity of our system but conda was unable to resolve the dependencies when creating an environment in the stepwise manner presented by the authors.

3) Incorrectly formatted VCF: The biggest concern was that we could not parse the VCF output using conventional tools such as bedtools intersect. The records as written are identified as being invalid.

Using the vcf_validator tool (from EBI) we see the following report for an example VCF file generated by DeBreak:

```
"According to the VCF specification, the input file is not valid
Error: INFO MAPQ does not match the meta specification Type=Integer (not in integer format). This occurs
795 time(s), first time in line 23.
Warning: A valid 'reference' entry is not listed in the meta section. This occurs 1 time(s), first time in line
23."
```

To understand this better, we did investigate the code. Overall the code is poorly documented and relies heavily on manual execution of tasks including compiling commands to run using os.system as well as manual writing of VCF files - the authors should consider using a library such as pysam to handle these functions to ensure compatibility. Essentially, the code could be significantly improved for both readability and speed. This is not essential for publication but it will be important for those seeking to use the tool in the future.

Ensuring the VCF file is the correct format is essential.

4) Tool versioning: With respect to the manuscript itself, I found the text clear and easy to interpret. The authors should specify the benchmark software versions used for cuteSV, sniffles etc. This is particularly important as sniffles has recently been updated to version 2 and I assume the work presented here is with

respect to an earlier version. Similarly cuteSV has been updated many times since its publication. For the record, neither cuteSV (1.0.13) or Sniffles (vs2) could detect the known breakpoint in our sample- which was detected by DeBreak.

5) Overclaims: Some of the language in the manuscript requires moderation. The opening of the introduction argues that SVs play the major role in all human genomic variation. But my interpretation of the word "genomic" would exclude single nucleotide variation and so all one is left with is SVs. The following sentence argues that SVs contribute more diversity than any other type of variant - are the authors talking at the population level? The individual level? Arguably we have a far better understanding of single nucleotide variants than we do SVs at the population scale. I agree with the overall point that the authors are making but they could moderate their claims without diluting the message or significance of the manuscript.

6) Comparison with assembled genomes: The authors should also consider that the work of Chaisson et al on analysing SVs in CHM13 can now be directly compared with a completed genome.

7) Table Details: In Table S1 (genotyping accuracy), the authors show poorer performance for debreak in PacBio HiFi reads than CLR. This is surprising - every other tool improves its performance as the read quality improves. The authors do not comment on this but it is very interesting - do they have an explanation for why this might be? Is DeBreak somehow optimised for noisier reads? This also contrasts with the F1 scores presented in table S2 - where HiFi does result in improved performance.

Reviewer #2 (Remarks to the Author: Reproducibility):

My comments in the field above address some reproducibility issues. In essence detailed comparisons require a correctly formatted VCF file.

Reviewers #3-4 (Remarks to the Author: Overall significance):

Chen et al. present DeBreak, a new method for the detection of structural variants (SVs) from third-generation long-read sequencing data. Overall, the paper is well-written and clear and the presented experiments demonstrate the performance of the new algorithm, which is benchmarked against three other algorithms (Sniffles, pbsv and cuteSV) using simulated long-read data and Nanopore and PacBio long-read data from a real human genomes, including HG002. Structural variant discovery is an important problem, and DeBreak is a valuable addition to the bioinformatics toolchain for SVs. The algorithm's ability to accurately determine the breakpoints of SVs at the sequence level has, besides generating biological insight, the advantage that it enables an integration of the SV calls into the reference panels used for downstream short-read-based SV genotypers (e.g. ParaGraph or potentially PanGenie).

Overall, we are positive about the publication of this paper. There are, however, important points that should be addressed prior to publication.

Major:

- Abstract:

a) The last sentence "DeBreak also demonstrates excellent performance in supplementing whole-genome assembly methods." should be toned down - it is unclear based on the presented results whether the DeBreak-unique calls (compared to assembly-based methods) are true- or false-positive calls.

- Simulations:

a) Simulations are always a good starting point, but the simulation approach chosen by the authors is simplistic. Structural variants tend to emerge around areas of existing sequence homology, i.e. not randomly, but in more repeat-rich / difficult-to-analyze regions of the human genome. An improved or second simulation experiment could be constructed e.g. by varying the copy number of existing repeats in the human genome. Alternatively, the simulations part could be left as it is, but its limitations would have to be made more clear in the section describing the results and in the Discussion.

- b) In the reduced coverage simulations part of the results, it would be important to assess the effect of coverage on SV breakpoint detection accuracy.
- c) DeBreak includes a specific component for the detection of long insertions. It would be important to better understand the specific contribution of this component to the algorithm's overall ability to detect insertions of different lengths (i.e., conditional on insertion length). Also, the paper includes a real-data evaluation of DeBreak's performance on insertions conditional on insertion length (see Figure 2); but even in the 50kb bin, DeBreak still achieves an F1 score of about 0.75. It would be important to better understand from which insertion size onwards the performance of DeBreak really starts breaking down; this could be assessed using simulations.

- HG002 results:

- a) The authors highlight the importance of multi-allelic copy number variations (line 139) - what exactly is meant by "multi-allelic" here? Multi-allelic in one genome or in the human population? Also, if truth data are available multi-allelic copy number variants in HG002, it would be important to measure the accuracy of DeBreak's multi-allelic calls against these (genotype accuracy and breakpoint accuracy).
- b) Downsampling: these results are very relevant; it would be important, however, to also understand the effect of reduced coverage on breakpoint detection accuracy.
- c) Line 174: "Taken together, these results highlight that DeBreak can accurately identify different types of SVs with precise breakpoints in real human genomes.". This statement should be explicitly limited to insertions and deletions (which are the types of SV calls that were validated here).

- HGSVC results:

- a) It would be important to also specify and discuss recall and precision of the different SV detection methods (Table S2 only specifies F1 scores). An additional table with these metrics could be added in which all evaluated samples (of one sequencing technology type) are combined.
- b) Why was Sniffles not applied to these samples?
- c) Why is no evaluation of SV genotype accuracy carried out on this dataset?

- Discussion:

- a) It would be good to include a brief section on the effect of the sequencing read data type (CLR, HiFi, Nanopore) on accuracy.
- b) While DeBreak can call insertions, deletions, inversions, and translocations, the empirical validation focuses on insertions and deletions (due to the availability of truth SV data). A brief paragraph discussing this and highlighting the fact that additional empirical validation of the algorithm's performance on inversions and translocations would be desirable should be added to the Discussion.

- Methods:

- a) Overall the Methods section is too vague and not specific enough.
- b) Raw SV signal detection: Please be more specific about how exactly "raw SV signals" are detected based on the alignments. Please provide details on the density-based clustering of SV signals.
- c) It is unclear to us what happens to translocations during the SV detection and clustering phase; in particular, it is unclear to us how translocations between different chromosomes are detected / handled (as chromosomes are processed independently).
- d) Large insertion detection via local assembly: How exactly is "enriched clipped alignment" defined? It is not clear to us how this step differs from the main raw SV signal detection step - which is also followed by a POA-based reassembly step? Also, it would seem to us that one key problem with long insertions is that the insertion may be longer than the read length - if we understand the approach of DeBreak correctly, however, the local re-assembly will only include reads that extend into the areas outside of the insertion (and thus miss the "middle region" of long insertions)? Last, but not least, how exactly is the local de novo assembly step carried out (e.g. using an external long-read assembly algorithm)?

Minor:

- General points:

- a) PBHoney was mentioned in the introduction but is not evaluated - could the authors explicitly comment on why this is (or alternatively add it to the evaluations)?

- Simulations:

- a) There are 3 simulated datasets with varying read lengths, but Table 1 lumps them together - it would be good to see results broken down by read length (e.g. in the Supplement) to better understand potential impacts of read length. How exactly were translocations modeled (inter-chromosomal or across chromosomes)?
- b) Adding a brief explanation where the different SV length simulation peaks in Figure S1 come from to the figure's legend would be helpful.
- c) Figure S3b seems to show a systematic bias with respect to the breakpoint accuracy of DeBreak-determined duplications - could the authors comment on that?
- d) Also, the red bars in Figure S3b are kind of hard to identify (apart from the outliers, see previous point) - consider modifying the graph to enable a better distinction between the variant types?
- e) The simulations could be extended by a simulated Nanopore dataset; or a discussion point could be added explaining why the results of a Nanopore simulation would be expected to be similar to the PacBio simulation results.
- f) Line 118, "81.33% of SVs with ± 1 bp shift around the SV breakpoint": This statistic include correctly determind (= 0bp shift) breakpoints, correct? Phrasing this as "within 1bp of the true SV breakpoint" may be more clear.

- HG002 results:

- a) It is not exactly clear to us how genotype accuracy is calculated here. What are the possible "genotypes" in the truth set - just 0, 1 or 2, or does the truth also include copy-number-variant genotypes? Also, is genotyping accuracy calculated on the respective algorithm's call set, or on the complete call set (both ways to evaluate genotyping accuracy are important and should be included).
- b) Please also specify ± 1 bp breakpoint accuracy for all algorithms in the text (to make results fully comparable to the simulation experiment).
- c) Figure 2: Please make the utilized read data type (CLR, HiFi, Nanopore) more explicit in the figure and the legend.

- HGSVC results:

- a) Please also specify ± 1 bp breakpoint accuracy for all algorithms in the text (to make results fully comparable to the simulation experiment), and specify breakpoint accuracy as % of calls made
- b) It would be interesting (though not essential) to see results for multi-allelic copy number variations here as well.
- c) Line 207, "suggesting that alignment-based SV callers identified SV calls more consistently.": More consistently than assembly-based methods? As only one assembly-based data source (PAV) is considered here, this claim does not seem to follow.

- Cancer genome results:

- a) A brief description of the input data (technology type, coverage, read lengths) in the main text would be very helpful.
- b) Would it be possible to construct an assembly-based "truth set" here as well?
- c) The fusion gene-specific validation approach leveraging IsoSeq data is very interesting. Could the same approach be applied to the results of the other SV calling methods (to the extent that they are capable of calling translocations) as well? (We would view this as very interesting, though not essential)

- Methods:

- a) Explicitly mentioning the multi-allele step of the workflow may be helpful.
- b) Line 312: "The density of SV raw signal is computed for each position on the chromosome (Supp fig)" -> Figure number missing.
- c) Section 3.33 only mentions the PacBio CLR dataset, omitting the Nanopore and HiFi datasets (which are also downsampled).
- d) Line 342: Please define when exactly a read is counted as "supporting"

Reviewer 1:

Chen et al described DeBreak, a new structural variation (SV) caller for long sequence reads. DeBreak differs from the existing SV callers in its use of local reassembly, which I think is the right direction. The authors show that DeBreak outperforms other popular SV callers on both simulated and real datasets.

Response: We appreciate the reviewer's scrutiny of our work and thoughtful suggestions for improving the manuscript.

Major comments:

1) Please make DeBreak support python3. Python2 has retired for more than a year. Users would question the long-term commitment to DeBreak if they see it support python2 only. Python2-only also makes it difficult for others to contribute to DeBreak as there will be fewer Python2 programmers in future.

This point was also raised by Referee #2, and should be addressed for further consideration at *Nature Communications* and *Communications Biology*.

Response: Thanks for the suggestion about updating Python version. We understand that Python2 is now out of date and many developers have stopped updating packages for Python2. We have updated DeBreak to version 1.3, and it supports both Python2 and Python3 now.

2) It would be good to stratify the result by repeat classes. For example, what is the accuracy for ALUs, LINE1s, SVAs, STRs, VNTRs and non-repeats? I predict that the accuracy of every caller will be near perfect for ALUs and LINE1s and will drop a lot in VNTRs. It is rare to see such stratification in SV caller papers, but I think this is an important analysis and is likely to benefit this manuscript. The authors can run RepeatMasker/TRF on the longest allele to annotate repeats.

This point should be addressed for further consideration at *Nature Communications* and *Communications Biology*.

Response: Thanks for the suggestion. We have annotated the SVs in HG002 high-confidence SV callset and the SV callsets from DeBreak, pbsv, and cuteSV using RepeatMasker, respectively. Sniffles did not report sequences for alternative alleles in the VCF file and thus was not included in this experiment. We then evaluated the SV discovery accuracy for three SV callers in each repeat category.

As the reviewer predicted, overall, all three SV callers achieved relatively higher accuracy in SINE (including Alu elements), LINE (including LINE1), DNA, and LTR regions, but relatively lower accuracy in Satellite (VNTRs) regions (**Fig. S11**). Out of the 10 annotated repeat classes, DeBreak achieved the highest accuracy in 9 repeat classes using PacBio CLR data, 5 repeat classes using PacBio HiFi data, and 10 classes using Nanopore data, suggesting a higher accuracy of DeBreak in resolving repeat-associated SVs.

Figure S11 SV discovery accuracy in different repeat types in HG002. F1 score of SV discovery in high-confidence regions of HG002 using PacBio CLR, HiFi and Nanopore data. The repeat type was annotated with RepeatMasker using sequences of longest allele for each SV. LC, low-complexity.

3) I am a little concerned with the low consistency between DeBreak and PAV. Could the authors compare PAV HG002 calls to GIAB? It would be important to understand the accuracy of PAV. Another option is to use dipcall for assembly-based SV calling. Dipcall is known to agree with GIAB well. Its accuracy is lower than read-based SV calls mostly due to different variant representations. Note that dipcall also generates confident regions like GIAB.

This point should be addressed for further consideration at *Nature Communications and Communications Biology*.

Response: Thanks for raising this concern. We compared the callset of PAV from Audano et al. 2021 Science for HG002 (NA24385) HiFi data to the GIAB benchmark callset. The VCF file of PAV was based on hg38, so we used LiftOver to convert the coordinates to hg19 (22,551 SVs were successfully converted, and 537 SVs were unmapped). When restricted to the high-confidence regions suggested by GIAB, we observed high consistency between PAV and GIAB, with an F1 score of 96.50% for DEL and 93.69% for INS (**Table 2**). Such high accuracy is expected in the high-confidence regions, as most of the differences between DeBreak and PAV were located near telomere and centromere regions as shown in **Fig. 3d** and **Fig. S18**.

Table 2 SV discovery accuracy on HG002

	DeBreak			Sniffles			pbsv			cuteSV			PAV		
	Rec	Pre	F1	Rec	Pre	F1	Rec	Pre	F1	Rec	Pre	F1	Rec	Pre	F1
Deletion															
CLR	97.73	96.48	97.10	95.14	96.69	95.91	95.28	96.21	95.74	97.31	94.18	95.72	-	-	-
HiFi	98.16	95.11	96.61	97.45	91.67	94.47	96.74	94.88	95.80	97.71	93.33	95.47	96.60	96.40	96.50
Nano	98.40	95.07	96.71	96.29	94.62	95.45	97.40	81.08	88.50	98.18	89.27	93.51	-	-	-
Insertion															
CLR	97.15	93.36	95.22	88.38	89.58	88.98	93.22	83.43	88.05	95.40	81.67	88.00	-	-	-
HiFi	97.26	92.84	95.00	90.90	87.79	89.32	97.41	80.42	88.10	96.64	89.88	93.14	96.18	91.32	93.69
Nano	97.46	93.91	95.65	90.57	90.01	90.29	95.07	85.60	90.09	96.99	89.27	92.97	-	-	-

The unit for recall, precision, and F1 score is %. The highest recall, precision, and F1 score among four tested SV callers are marked in bold.

Rec = Recall. Pre = Precision. F1 = F1 score. Nano, Nanopore. -, Not applicable.

We have also applied Dipcall on the six HGSVC samples using haplotype-resolved assemblies provided by HGSVC group. Using the Dipcall's callset as "ground truth", we benchmarked the accuracy of four alignment-based SV callers on these samples. In general, we observed recall between 56% to 76% and precision between 74% to 83% (**Fig. R1b**), which was similar compared to PAV callset (**Fig. R1a**). When restricted to the confident regions suggested by Dipcall, the recall was reduced in general (54-67%) while the precision was largely increased (87-96%) (**Fig. R1c**), and the F1 scores were still at similar levels (67-79%).

We then performed a three-way comparison of SV callsets of DeBreak, PAV, and Dipcall (**Fig. R1d**). A large proportion of the SVs were reported by all three callers. Because the callsets of PAV have been curated to remove the low-confidence SV calls in the segmental duplications and misclustered contigs (Ebert et. al., 2021), there were fewer PAV-unique SVs in **Fig. R1d**. We decided to keep the PAV callsets as "ground truth" in assembly-based SV discovery in the Results section.

Figure R1 SV discovery accuracy compared to PAV and Dipcall callsets. **a** SV discovery accuracy of four alignment-based SV callers compared to PAV callset in six HGSC samples. **b,c** SV discovery accuracy of four alignment-based SV callers compared to Dipcall callset in whole genome (**b**) and in confident regions suggested by Dipcall (**c**). **d** Venn plot showing overlapping of deletions and insertions reported by DeBreak, PAV, and Dipcall. Number of SV in each category was labeled in Venn plots.

Minor comments:

4) What assembler is used for local assembly? Is it wtdbg2?

Several referees also commented on some confusion about specific tools or versions of software. For the sake of reproducibility, please carefully expand on the Methods, and refer to the Open Research Evaluation at the bottom of this document for further guidance.

Response: Thanks for the question. Yes, wtdbg2 was used for local assembly as it accepts long reads from all platforms and is much faster than other long-read assemblers based on our evaluation. We believe speed is important here, as there may be many candidate insertions sites throughout the genome. And thanks for the editor's suggestion. We have updated the Methods section to make it clearer.

5) Does DeBreak assemble all reads mapped to a candidate SV, or only assemble reads that contains the SV?

Response: Thanks for the question. DeBreak only assembles reads containing the SV signal to obtain more accurate consensus sequences. This approach is critical, especially for heterozygous SVs and SVs located in repeats such as segmental duplications. We have updated the Methods section to make it clearer.

6) What is the tool and the command line for comparing SV callsets? Is it truvari?

Response: Thanks for the question. We did not use truvari. Following the HGSC callsets merging guidelines, we used a custom Python script for SV callset comparison. Two SV events are considered as a match when 1) they have same SV type, 2) left breakpoints are located within 1000bp, 3) SV size ratio is larger than 0.5 and

smaller than 2. We have made the custom Python scripts used in our manuscript available at https://github.com/Maggi-Chen/DB_code.

7) The last two pages on both "Supplementary file"s are not properly formatted. I guess this is generated by PDF printing. It would be good to have Excel files instead as it is difficult to derive a text file from PDF.

Response: Thanks for raising this issue. In the two supplementary files, we listed the information about PCR primers used for SV validation and gene fusions reported by DeBreak in SKBR3 cell line. Both files were submitted as separate Excel files as they both have too many columns do not readily fit in a text file. The formatting was probably altered during the file merging and PDF printing process through the submission system.

Reviewer 2:

Firstly my apologies to the authors for a slow review.

I have attempted to run DeBreak on some samples and have found it to be a useful tool. I hope that this tool will be used in future. However if the authors wish it to be used widely it will require some tuning. We were able to identify known events in samples using DeBreak that were not identified in either cuteSV or sniffles (VERSION 2). This is useful and I have already added DeBreak to my suite of tools to use to look for SVs.

As I state below, I am already investigating using DeBreak to look at SVs alongside tools such as CuteSV and Sniffles. The analysis of SVs detected using long read technology is of huge interest and the tools available to do so are still maturing. DeBreak is a worthy tool in this suite of methods.

Response: Thanks for testing DeBreak and for the positive comments about our work. This is really encouraging!

Major comments:

1) Installation 1: I do have several comments about the code and its implementation that have been challenging. Firstly, the code is written in python 2.7 - now end of life (not supported). Really the tool should be updated to a current version of python. Similarly, the dependencies required for installing are old with all having been updated in the last two years versus the tested versions described.

This point was also raised by Referee #1, and should be addressed for further consideration at *Nature Communications and Communications Biology*.

Response: Thanks for raising the concern. We have updated DeBreak to version 1.3, which supports Python3 and newer versions of dependencies.

2) Installation 2: We also could not get the code to install using the suggested conda instructions - instead we had to install in a specific environment file (see below for the yaml file we created to make an environment). This may have been a peculiarity of our system but conda was unable to resolve the dependencies when creating an environment in the stepwise manner presented by the authors.

Response: Thanks for pointing this out. To simplify installation process, we have uploaded our package to the bioconda channel of Anaconda. It now can be installed with 'conda install -c bioconda debreak' without the need to install dependencies manually.

3) Incorrectly formatted VCF: The biggest concern was that we could not parse the VCF output using conventional tools such as bedtools intersect. The records as written are identified as being invalid.

This point should be addressed for further consideration at *Nature Communications*.

Using the vcf_validator tool (from EBI) we see the following report for an example VCF file generated by DeBreak:

"According to the VCF specification, the input file is not valid Error: INFO MAPQ does not match the meta specification Type=Integer (not in integer format). This occurs 795 time(s), first time in line 23.

Warning: A valid 'reference' entry is not listed in the meta section. This occurs 1 time(s), first time in line 23."

To understand this better, we did investigate the code. Overall the code is poorly documented and relies heavily on manual execution of tasks including compiling commands to run using os.system as well as manual writing of VCF files - the authors should consider using a library such as pysam to handle these functions to ensure compatibility. Essentially, the code could be significantly improved for both readability and speed. This is not essential for publication but it will be important for those seeking to use the tool in the future.

Ensuring the VCF file is the correct format is essential.

We strongly encourage you to better annotate the code for further consideration at *Nature Communications and Communications Biology*.

Response: Thanks for noticing this VCF format error. We have updated DeBreak to version 1.3, and it now uses package 'pysam' to write its VCF output. Currently, the VCF file generated by DeBreak can be validated with vcf_validator (**Fig. R2**). The header tag 'reference' was not present in VCF as it is not present in the input

BAM file (generated by minimap2 by default). We have also annotated the code of DeBreak to improve readability.

```
(pack) [maggic@login004 new_sim_pacbio_repl]$ vcf-validator -u debreak.vcf
The header tag 'reference' not present. (Not required but highly recommended.)

-----
Summary:
  1 errors total

  1 .. The header tag 'reference' not present. (Not required but highly recommended.)
```

Figure R2 Validation of DeBreak output VCF file with vcf-validator. VCF file was generated by DeBreak v1.3 on simulated dataset replicate 1 using PacBio-like reads.

4) Tool versioning: With respect to the manuscript itself, I found the text clear and easy to interpret. The authors should specify the benchmark software versions used for cuteSV, sniffles etc. This is particularly important as sniffles has recently been updated to version 2 and I assume the work presented here is with respect to an earlier version. Similarly cuteSV has been updated many times since its publication. For the record, neither cuteSV (1.0.13) or Sniffles (vs2) could detect the known breakpoint in our ample- which was detected by DeBreak.

Similar concerns were raised by other referees; please be sure to elaborate on the Methods section and clarify the rationale for each analytical step, as well as relevant versions of software.

Response: Thanks for the suggestion. The SV callsets benchmarked in the manuscript were generated with DeBreak v1.2, cuteSV v1.0.11, pbsv v2.6.2, and Sniffles v1.0.8. Sniffles used for comparison is still version 1 as version 2 was not officially released by the time we performed the experiments. We have added the specific version of each tool used in the Methods section.

5) Overclaims: Some of the language in the manuscript requires moderation. The opening of the introduction argues that SVs play the major role in all human genomic variation. But my interpretation of the word "genomic" would exclude single nucleotide variation and so all one is left with is SVs. The following sentence argues that SVs contribute more diversity than any other type of variant - are the author talking at the population level? The individual level? Arguably we have a far better understanding of single nucleotide variants than we do SVs at the population scale. I agree with the overall point that the authors are making but they could moderate their claims without diluting the message or significance of the manuscript.

This point was also raised by the other reviewers, please carefully qualify the manuscript to avoid any overstatements.

Response: Thanks for pointing this out. The argument that SVs contribute more diversity than any other type of variant was referred to Chaisson MJP et al. 2019 Nature Communications paper – first sentence of their Introduction. In that paper, 27k SVs (≥ 50 bp) and 818k indels (< 50 bp) per human genome were discovered. As calculated in Audano P et al. 2019 Cell, on average, SVs affect ~ 11 Mbp per human genome. We interpreted the diversity at the nucleotide level between two human genomes in that paper was referring to total bases of all SVs in size. We know that the diversity/polymorphism of SNVs in humans is approximately 0.1%, which is ~ 3 -4 million per individual. We appreciate the reviewer's suggestion to moderate the claim without diluting the significance of the work. Therefore, we have revised the Introduction section about the importance of SVs in human genome to: "Structural variations (SVs), or genomic rearrangements, including insertions, deletions, inversions, duplications, translocations, and complex forms of multiple events, contribute a large proportion of genetic variations in many species. In humans, SVs affect larger genomic regions in size than any other type of variants and play a pathogenic role in a wide range of genetic disorders."

6) Comparison with assembled genomes: The authors should also consider that the work of Chaisson et al on analysing SVs in CHM13 can now be directly compared with a completed genome.

This point could simply be mentioned as a future direction for *Communications Biology*.

Response: Thanks for the suggestion. We have downloaded the telomere-to-telomere assemblies of CHM13 and generated an assembly-based SV callset using Dipcall. Since CHM13 is isogenic, we believe Dipcall results should be of high confidence. PacBio CLR, HiFi, and Nanopore data of CHM13 were downloaded and mapped to the human reference genome. DeBreak, Sniffles, pbsv, and cuteSV were applied to generate alignment-based SV callsets. Four alignment-based SV callsets were compared to the assembly-based callset within the high-confidence regions suggested by Dipcall.

In all three data types, DeBreak showed the highest consistency with the assembly-based callset for both deletions and insertions (Table S8). We observed a higher F1 score in Nanopore than CLR and HiFi dataset, likely owing to the large differences in sequencing depth of the three datasets (~70x for CLR, ~55x for HiFi, and ~126x for Nanopore). On CHM13 genome, all SVs should be homozygous. We then benchmarked the genotyping accuracy of the four alignment-based SV callers. DeBreak achieved the highest genotyping accuracy in PacBio HiFi and Nanopore datasets, and the genotyping accuracy in the PacBio CLR dataset is slightly less than cuteSV (Table S9). We have updated the Results and Methods section for SV benchmark in the CHM13 cell line.

Table S8 SV discovery accuracy in CHM13

	Deletion			Insertion			Total		
	Recall	Precision	F1	Recall	Precision	F1	Recall	Precision	F1
CLR									
DeBreak	78.24	87.49	82.61	75.23	85.53	80.05	76.36	86.27	81.01
Sniffles	74.39	86.61	80.03	64.18	86.32	73.62	68.00	86.44	76.12
pbsv	77.29	84.22	80.61	78.24	66.01	71.61	77.89	71.77	74.70
cuteSV	78.40	86.45	82.23	75.59	80.74	78.08	76.64	82.83	79.62
HiFi									
DeBreak	79.43	83.55	81.44	76.72	87.72	81.85	77.73	86.08	81.69
Sniffles	75.07	83.96	79.27	64.80	82.59	72.62	68.65	83.14	75.20
pbsv	76.20	83.25	79.57	79.93	61.80	69.70	78.53	68.17	72.99
cuteSV	81.01	79.56	80.28	79.29	81.29	80.28	79.93	80.63	80.28
Nanopore									
DeBreak	81.30	80.67	80.99	78.89	85.91	82.25	79.79	83.84	81.76
Sniffles	77.46	78.35	77.91	67.46	78.68	72.64	71.20	78.55	74.70
pbsv	79.82	54.24	64.59	82.07	65.97	73.14	81.23	61.11	69.75
cuteSV	82.80	71.85	76.94	79.66	77.42	78.53	80.83	75.19	77.91

SV discovery accuracy was evaluated with the assembly-based SV callset as the ground truth. The highest F1 score in each SV type are shown in bold. The unit of recall, precision, and F1 score is %.

Table S9 SV genotyping accuracy in CHM13

	CLR	HiFi	Nanopore
DeBreak	77.00	86.02	78.74
Sniffles	32.03	26.19	48.73
pbsv	62.65	81.31	59.33
cuteSV	77.33	74.36	61.50

SV genotyping accuracy was evaluated with only 'GT=1/1' as correct genotype. The highest genotyping accuracy in each data type are shown in bold. The unit of genotyping accuracy is %.

7) Table Details: In Table S1 (genotyping accuracy), the authors show poorer performance for debreak in PacBio HiFi reads than CLR. This is surprising - every other tool improves its performance as the read quality improves. The authors do not comment on this but it is very interesting - do they have an explanation for why this might be? Is DeBreak somehow optimised for noisier reads? This also contrasts with the F1 scores presented in table S2 - where HiFi does result in improved performance.

This point should be addressed for further consideration at *Nature Communications and Communications Biology*.

Response: Thanks for pointing this out. Higher base accuracy of HiFi data does benefit SV discovery and breakpoint accuracy. However, sequencing depth has larger impact on genotyping accuracy of DeBreak than base accuracy. We performed down-sampling for PacBio CLR, HiFi and Nanopore datasets and assessed the genotyping accuracy in depth ranging from 10x to 100x. In general, genotyping accuracy increased as the sequencing depth increased (**Fig. S14**). Input data type had minor influence on the genotyping accuracy, as three data types had similar accuracy at each depth. In **Table S1**, the CLR dataset was ~70x, while the HiFi dataset was ~50x. Based on the results shown in **Fig. S14**, genotyping accuracy in HiFi at 50x was lower than in CLR at 70x, so we observed lower accuracy in HiFi than CLR in **Table S1**. We have added the genotyping accuracy benchmark in downsampled datasets in the Results section.

Figure S14 Genotyping accuracy in down-sampled datasets in HG002. The PacBio CLR dataset was downsampled from 10x to 70x. The PacBio HiFi and Nanopore datasets were downsampled from 10x to 100x.

Reviewer 3-4:

Chen et al. present DeBreak, a new method for the detection of structural variants (SVs) from third-generation long-read sequencing data. Overall, the paper is well-written and clear and the presented experiments demonstrate the performance of the new algorithm, which is benchmarked against three other algorithms (Sniffles, pbsv and cuteSV) using simulated long-read data and Nanopore and PacBio long-read data from a real human genomes, including HG002. Structural variant discovery is an important problem, and DeBreak is a valuable addition to the bioinformatics toolchain for SVs. The algorithm's ability to accurately determine the breakpoints of SVs at the sequence level has, besides generating biological insight, the advantage that it enables an integration of the SV calls into the reference panels used for downstream short-read-based SV genotypers (e.g. ParaGraph or potentially PanGenie). Overall, we are positive about the publication of this paper. There are, however, important points that should be addressed prior to publication.

Response: We really appreciate the reviewer's approbation and suggestions on our work. We have addressed the reviewer's concerns by performing additional experiments. These comments are very helpful for the improvement of DeBreak and the manuscript.

Major:

- Abstract:

a) The last sentence "DeBreak also demonstrates excellent performance in supplementing whole-genome assembly methods." should be toned down – it is unclear based on the presented results whether the DeBreak-unique calls (compared to assembly-based methods) are true- or false-positive calls.

Response: Thanks for pointing this out. We have revised this sentence to "DeBreak can also be used for supplementing whole-genome assembly-based SV discovery." to avoid overclaim.

- Simulations:

a) Simulations are always a good starting point, but the simulation approach chosen by the authors is simplistic. Structural variants tend to emerge around areas of existing sequence homology, i.e. not randomly, but in more repeat-rich / difficult-to-analyze regions of the human genome. An improved or second simulation experiment could be constructed e.g. by varying the copy number of existing repeats in the human genome. Alternatively, the simulations part could be left as it is, but its limitations would have to be made more clear in the section describing the results and in the Discussion.

It would be essential that you address this point with additional simulations and analysis for further consideration at *Nature Communications*. This point could be addressed via discussion of limitations for *Communications Biology*.

Response: Thanks for the advice. We have added another simulation experiment of copy number variation of repeats. We randomly picked 10,000 existing repeats (with size between 50bp to 20kbp) in the repeat annotation from RepeatMasker and then modified human reference genome at these repeat regions by removing or adding a copy of the repeat unit to mimic SVs in repetitive regions. 5,000 SVs were assigned to both haplotypes as homozygous SVs, and the other 5,000 SVs were assigned to only one haplotype as heterozygous SVs. PacBio-like and Nanopore-like sequencing reads were simulated using pbsim and Badread with average read length of 10kbp, 15kbp, and 20kbp, similar as the previous simulation experiment. Sequencing reads were aligned to human reference genome GRCh38. DeBreak, Sniffles, pbsv, and cuteSV were applied on the read alignment files to call SVs. A series of "--min_supp" were provided to Sniffles and cuteSV, and the SV callsets with highest accuracy were used for comparison.

DeBreak achieved accuracy of 98.67% and 97.71% using PacBio and Nanopore data respectively, which was higher than the other three SV callers (**Table S2**). In both deletion and insertion detection, DeBreak achieved highest F1 score among four tested SV callers using both PacBio and Nanopore data. When compared to the previous randomly simulated SVs, SV discovery accuracy for repeat-associated SVs was slightly lower for both deletions and insertions. We have updated the Results and Methods section regarding the repeat-associated SV simulation.

Table S2 SV discovery accuracy of SV involving repeats

Type	Deletion			Insertion			Total		
	Recall	Precision	F1	Recall	Precision	F1	Recall	Precision	F1
PacBio									
DeBreak	99.03	99.86	99.44	97.69	98.12	97.91	98.36	98.99	98.67
Sniffles	97.72	99.86	98.78	95.19	97.85	96.49	96.46	98.85	97.64
Pbsv	98.13	99.97	99.04	88.95	98.26	93.37	93.54	99.15	96.26
cuteSV	97.97	99.97	98.96	94.77	97.06	95.89	96.37	98.52	97.43
Nanopore									
DeBreak	98.12	98.51	98.31	97.35	96.89	97.12	97.73	97.69	97.71
Sniffles	97.65	97.98	97.81	93.01	98.21	95.52	95.33	98.09	96.68
Pbsv	99.13	97.25	98.18	92.60	97.95	95.20	95.87	97.58	96.72
cuteSV	98.31	97.95	98.13	93.87	97.60	95.70	96.09	97.78	96.93

Averages for three replicates (10kbp, 15kbp,20kbp). The unit for recall, precision, and F1 score is %. The highest recall, precision, and F1 score for each category are shown in bold.

b) In the reduced coverage simulations part of the results, it would be important to assess the effect of coverage on SV breakpoint detection accuracy.

This point (and point c, below) would be necessary for further consideration at *Nature Communications* and *Communications Biology*.

Response: Thanks for the suggestion. We have assessed the breakpoint accuracy of DeBreak in down-sampled simulation datasets. The original 50x datasets were down-sampled from 5x to 45x. Overall, the breakpoint accuracy was improved when sequencing depth increased in both PacBio and Nanopore simulations (Fig. S8). The breakpoint accuracy of DeBreak saturated at approximately 20x. We did not observe large differences between PacBio CLR and Nanopore datasets, as both simulations had high error rates. We have added a supplementary figure and updated the Results section in the manuscript.

Figure S8 SV breakpoint accuracy in down-sampled simulated datasets. Number of detected SVs with exact breakpoint (solid line) and shift ≤ 1 bp (dashed line) in three PacBio and Nanopore replicates.

c) DeBreak includes a specific component for the detection of long insertions. It would be important to better understand the specific contribution of this component to the algorithm's overall ability to detect insertions of different lengths (i.e., conditional on insertion length). Also, the paper includes a real-data evaluation of DeBreak's performance on insertions conditional on insertion length (see Figure 2); but even in the 50kb bin, DeBreak still achieves an F1 score of about 0.75. It would be important to better understand from which

insertion size onwards the performance of DeBreak really starts breaking down; this could be assessed using simulations.

Response: Thanks for the suggestion. We have added another simulation of ultra-large insertion detection to access the maximal insertion length detected by DeBreak. 1,000 insertions were simulated and embedded into Chromosome 1, with insertion size ranging from 5kbp to 100kbp. PacBio reads were simulated with average read length of 15kbp. We then applied four SV callers and benchmarked the recall at different size ranges. In general, INS detection recall dropped as the insertion size increased for each SV caller, and DeBreak achieved the highest recall in each size category (**Fig. S6**). For insertions that can be fully contained in the sequencing reads (5-10kbp), all SV callers detected INS with relatively high recall. When insertion size was similar to the read length (10-20kbp), the recall dropped dramatically for Sniffles, pbsv, and cuteSV, but DeBreak still achieved a recall over 75%. When insertion size exceeded sequencing reads (20k-30k), DeBreak identified 70% of homozygous INS and 40% of heterozygous INS, while the other three SV callers failed to detect any events. The maximal detectable insertion size of DeBreak was roughly twice of the average read length, as the recall dropped dramatically when insertions were longer than 30kbp.

Based on the simulation results, we showed that the maximal detectable insertion size of DeBreak was approximately twice of the average read length, which was expected considering the principle of large-insertion detection module. Insertions larger than 30kbp can hardly be detected, as no full-length contig can be constructed from local assembly using reads on both side of the breakpoint. In particular, we observed a higher recall for detecting homozygous insertions than heterozygous insertions at 20-30kbp, as homozygous SVs have more SV-containing reads for local *de novo* assembly and therefore have higher success rate in generating full-length contigs.

Figure S6 Large insertion detection in simulated datasets. Recall of insertion detection at different size ranges. The average length of sequencing reads was 15kbp. The maximal detectable insertion size is 10kbp for Sniffles, 20kbp for pbsv and cuteSV, and 30kbp for DeBreak.

- HG002 results:

a) The authors highlight the importance of multi-allelic copy number variations (line 139) - what exactly is meant by "multi-allelic" here? Multi-allelic in one genome or in the human population? Also, if truth data are available multi-allelic copy number variants in HG002, it would be important to measure the accuracy of DeBreak's multi-allelic calls against these (genotype accuracy and breakpoint accuracy).

Response: Thanks for raising this concern. The 'multi-allelic' SV here represents SVs with two different alleles in one genome (the sequenced sample) that are both different from the reference allele. For these SVs, there are at least three alleles in human population, so we used the multi-allelic term. This is an under-estimation of multi-allelic SVs in human populations. There is a chance of two non-reference segregating alleles occur in two individuals, as we have identified from HG002 samples (data not shown). In the benchmarked high-

confidence SV callset of HG002, we checked and found that all SVs had only one alternative allele. These SVs were detected from different platforms (Illumina, 10x genomics, PacBio, Bionano, etc.) and then merged as a high-confidence callset. Multiple alleles of the same SV (in the same region) were merged into one allele during this process. Thus, the truth SV set is not suitable for serving as ground truth for multi-allelic SVs.

When comparing mSVs identified by DeBreak to the truth SV callset, most mSVs (78/87, 89.66% in CLR; 49/53, 92.45% in HiFi; 74/90, 82.22% in Nanopore) had at least one of the two alleles matched with an SV event in the truth set, suggesting that these are true SVs. To further confirm the presence of the other allele, we checked the read alignments for these mSVs. There are SV raw signals from both alleles (two groups of raw signals with distinct size) in the read alignments of all three independent datasets (**Fig. S13**), suggesting

Figure S13 Example mSV in HG002 high-confidence regions. Two alternative alleles were reported by DeBreak (238bp DEL and 68bp DEL) for this mSV, and one of the two alleles matched with the truth SV set (272bp DEL). In PacBio CLR, HiFi, and Nanopore datasets, raw signals of both sizes (~250bp and ~70bp) are present in read alignments in this region.

that there are more than one alternative alleles at this region and that the truth SV set only includes one of them. We have added a Supplementary figure and updated the Results section about mSV benchmarks in HG002.

b) Downsampling: these results are very relevant; it would be important, however, to also understand the effect of reduced coverage on breakpoint detection accuracy.

This is similar to *Simulations: Point B* noted above.

Response: Thanks for the advice. We have assessed the SV breakpoint accuracy in downsampled datasets of HG002. Overall, the SV breakpoint accuracy was improved when sequencing depth increased, as more SVs were identified with exact breakpoint and with ≤ 1 bp shift at higher sequencing depths (**Fig. S19**). DeBreak and pbsv significantly outperformed Sniffles and cuteSV in all three data types. In PacBio CLR datasets, DeBreak exceeded pbsv at 30x and reported most SVs with exact breakpoint and ≤ 1 bp shift among four SV callers when depth was higher than 30x. In PacBio HiFi and Nanopore datasets, DeBreak exceeded pbsv at 20x and reported more SVs with exact breakpoint locations. We have added a supplementary figure and updated the Results section about breakpoint accuracy in downsampled datasets.

Figure S19 Breakpoint accuracy in downsampled datasets in HG002. Number of SV detected with exact breakpoint and with ≤ 1 bp shift in breakpoints using PacBio CLR, HiFi, and Nanopore data at different sequencing depths. PacBio CLR data was downsampled from 10x to 70x. PacBio HiFi and Nanopore data were downsampled from 10x to 100x.

c) Line 174: "Taken together, these results highlight that DeBreak can accurately identify different types of SVs with precise breakpoints in real human genomes.". This statement should be explicitly limited to insertions and deletions (which are the types of SV calls that were validated here).

As previously stated, please carefully qualify any claims to avoid overstatements.

Response: Thanks for the advice. We have revised the conclusion to "Taken together, these results highlight that DeBreak can accurately identify two major types of SVs, deletions and insertions, with precise breakpoints in real human genomes." in the Results section.

- HGSVC results:

a) It would be important to also specify and discuss recall and precision of the different SV detection methods (Table S2 only specifies F1 scores). An additional table with these metrics could be added in which all evaluated samples (of one sequencing technology type) are combined.

Response: Thanks for the suggestion. We have included an additional supplementary table (Table S5) to list the recall and precision of deletion and insertion detection for three samples and combined (Total) for CLR and HiFi data. In both CLR and HiFi datasets, DeBreak achieved the highest recall for both deletion and insertion detection and achieved the highest precision for deletion, while Sniffles achieved slightly higher precision for insertions than DeBreak.

Table S5. SV discovery recall and precision compared to assembly-based SV callsets

	DeBreak				pbsv				cuteSV				Sniffles			
	R-D	P-D	R-I	P-I	R-D	P-D	R-I	P-I	R-D	P-D	R-I	P-I	R-D	P-D	R-I	P-I
CLR																
HG00096	79.46	81.62	78.08	79.11	77.24	80.25	75.64	79.94	78.44	74.67	65.40	74.95	75.27	81.29	61.67	80.79
HG01505	79.40	82.20	76.68	80.87	77.41	81.04	74.47	80.81	78.99	75.03	64.51	75.34	75.04	81.04	59.80	82.42
HG01596	77.68	82.79	75.63	76.03	76.04	80.66	73.89	70.41	77.44	73.58	63.59	74.75	71.21	84.47	59.53	76.14
Total	78.85	82.19	76.79	78.64	76.90	80.65	74.66	76.79	78.30	74.43	64.50	75.01	73.86	82.18	60.33	79.71
HiFi																
HG02818	81.44	82.36	76.54	85.04	78.40	78.50	73.41	81.13	80.36	76.59	60.19	80.97	75.94	80.18	58.66	87.12
HG03486	83.14	81.22	79.31	83.98	81.29	78.20	77.26	80.09	81.71	75.64	60.85	80.47	78.20	79.18	60.94	86.54
NA12878	81.80	79.80	78.58	84.99	75.91	78.78	72.78	80.71	79.19	76.45	59.90	81.81	72.65	78.30	57.98	87.07
Total	82.15	81.19	78.13	84.65	78.71	78.46	74.57	80.62	80.50	76.21	60.33	81.04	75.79	79.28	59.26	86.90

SV discovery accuracy was evaluated with the assembly-based SV callset as the ground truth. The highest recall and precision among four tested alignment-based SV callers is marked in bold. The unit of recall and precision is %.
R-D, recall for deletion. P-D, precision for deletion. R-I, recall for insertion. P-I, precision for insertion.

b) Why was Sniffles not applied to these samples?

This point should be addressed for further consideration at *Nature Communications*.

Response: Thanks for the question. We had initially selected DeBreak, pbsv, and cuteSV for comparison with assembly-based approach as these three alignment-based SV callers had higher accuracy in previous benchmark. We have now applied Sniffles to the six HGSVC samples for comparison as well.

When compared to assembly-based SV callset, DeBreak still achieved the highest accuracy (F1 score) among four tested tools in all six samples, while Sniffles showed the lowest accuracy in both CLR and HiFi samples (**Table S4**). In all samples, DeBreak identified SVs with a higher recall than the other three callers (**Fig. 3a**). We have further assessed the breakpoint accuracy of Sniffles and found that DeBreak achieved the best breakpoint accuracy among all tested callers (**Fig. 3b**). Sniffles and cuteSV reported much fewer SVs with exact breakpoints than DeBreak and pbsv, which was consistent with the findings in HG002. We have updated the Results section to add Sniffles' results into comparison.

Table S4 SV discovery accuracy compared to assembly-based SV callsets

Sample	DeBreak			pbsv			cuteSV			Sniffles		
	DEL	INS	Total	DEL	INS	Total	DEL	INS	Total	DEL	INS	Total
CLR												
HG00096	80.52	78.59	79.34	76.51	69.85	72.59	78.72	77.73	78.12	78.39	69.54	73.18
HG01505	80.77	78.72	79.52	76.96	69.51	72.57	79.18	77.51	78.16	78.03	69.00	72.73
HG01596	80.15	75.83	77.45	75.46	68.72	71.47	78.28	72.10	74.39	76.76	66.80	70.69
HiFi												
HG02818	81.90	80.57	81.12	78.43	69.05	73.19	78.45	77.08	77.65	77.75	65.73	71.01
HG03486	82.17	81.58	81.82	78.56	69.30	73.40	79.71	78.65	79.09	79.14	68.22	73.01
NA12878	80.78	81.66	81.31	77.80	69.17	72.87	77.32	76.54	76.85	73.64	63.88	68.00

SV discovery accuracy was evaluated with the assembly-based SV callset as the ground truth. The highest accuracy (F1 score) among four tested alignment-based SV callers is shown in bold in each sample. The unit of F1 score is %.

Figure 3 Alignment-based and assembly-based SV discovery. **a** SV discovery recall and precision of alignment-based SV callers when compared with the assembly-based SV callset. **b** SV breakpoint accuracy of DeBreak, pbsv, cuteSV, and Sniffles in PacBio CLR (top) and HiFi (bottom) datasets. **c** Venn diagram showing the overlap among four SV callsets. The number of SV events in each category is labeled within each section. **d** Distribution of PAV-unique and DeBreak-unique SV calls on chromosomes 1-5. Red boxes indicate positions of centromeres.

c) Why is no evaluation of SV genotype accuracy carried out on this dataset?

Response: Thanks for the suggestion. We have added benchmark for SV genotyping accuracy in HGSC samples. The SV genotyping accuracy was assessed with the assembly-based SV callset as ground truth. Among the 6 samples, cuteSV achieved the highest genotyping accuracy in 5 samples, and DeBreak achieved the highest accuracy in 1 sample (Table S6). While in homozygous SV genotyping based on CHM13 evaluation DeBreak performed better than cuteSV, there is roughly 2% difference between DeBreak and cuteSV based on HGSC samples. Since there is no real ground truth data, we cannot simply conclude the genotyping accuracy

of DeBreak is less than cuteSV. However, we noticed that all the SV callers were not performing well in term of genotyping – there is a room to improve genotyping results. We will work on improving the genotyping accuracy in future releases. We have added the results about genotyping accuracy into the Results section.

Table S6 SV genotyping accuracy in HGSVC samples

	DeBreak	pbsv	cuteSV	Sniffles
CLR				
HG00096	72.70	67.81	75.06	45.88
HG01505	73.18	67.82	75.82	46.13
HG01596	70.04	65.11	68.75	39.05
HiFi				
HG02818	72.62	72.91	74.91	45.45
HG03486	71.75	72.54	74.13	45.25
NA12878	71.68	70.43	73.52	39.06

SV genotyping accuracy was assessed with the assembly-based SV callset as the ground truth. The highest accuracy among four tested alignment-based SV callers is shown in bold for each sample. The unit of genotyping accuracy is %.

- Discussion:

a) It would be good to include a brief section on the effect of the sequencing read data type (CLR, HiFi, Nanopore) on accuracy.

Response: Thanks for the advice. We have added a paragraph in the Discussion section to discuss the effects of input data type, sequencing depth, and read length on SV discovery accuracy: “Several features of the input sequencing dataset have essential impact on SV discovery accuracy. Data type (sequencing platform) affects SV discovery accuracy and breakpoint accuracy. Based on our benchmarks and as expected, datasets with lower sequencing error rates often lead to better SV discovery accuracy and breakpoint accuracy than datasets with higher error rates at similar levels of sequencing depth. Sequencing depth also affects accuracy for SV discovery, breakpoint position, and genotyping. Sequencing read length can affect maximal size of detectable SVs, especially for insertion detection.”

b) While DeBreak can call insertions, deletions, inversions, and translocations, the empirical validation focuses on insertions and deletions (due to the availability of truth SV data). A brief paragraph discussing this and highlighting the fact that additional empirical validation of the algorithm's performance on inversions and translocations would be desirable should be added to the Discussion.

Response: Thanks for the suggestion. We have added the following paragraph to the Discussion section to state the limitation of current benchmark in real datasets: “Due to the limited availability of ground-truth SV sets, DeBreak was benchmarked for insertion and deletion discovery in HG002 and HGSVC samples, but not for duplication, inversion, or translocation. Further validation of SV discovery accuracy on these SV types would be desirable and will help improve DeBreak’s performance if comprehensive high-confidence truth SV sets become more readily available.”

- Methods:

a) Overall the Methods section is too vague and not specific enough.

Response: Thanks for pointing this out. We have substantially revised the Methods section to add more details as suggested by the other reviewers as well.

b) Raw SV signal detection: Please be more specific about how exactly "raw SV signals" are detected based on the alignments. Please provide details on the density-based clustering of SV signals.

Response: Thanks for the suggestions. We have added a supplementary figure to show how SV raw signals are detected from read alignments. In general, raw signals are inferred from two approaches: within-alignment and split-read alignments. Smaller deletions and insertions can be directly inferred from single alignment of a read based on the information in the CIGAR string in BAM file (**Fig. S1a**). Larger deletions and insertions and other three types of SVs are inferred from two or more alignments of the same read. Each SV type has its own read alignment pattern (**Fig. S1b**).

a Within-alignment

b Split-read alignments

Figure S1 SV raw signal detection of DeBreak. **a** Deletion and insertion can be directly inferred within a single-read alignment. **b** Larger deletion and insertion, duplication, inversion, and translocation can be inferred from split-read alignments based on the location and orientation of two alignment segments. Alignments with distinct colors represent separate alignments of the same sequencing read.

For SV raw signal clustering, we have also included a new supplementary figure to demonstrate the clustering process (**Fig. S2**). All SV raw signals from same chromosome with same SV type are sorted based on the coordinates. The density of raw signal is calculated for each position on the reference genome. DeBreak then scans the density for peaks above an auto-adjusted threshold, which was determined based on the sequencing depth of input dataset. For each peak, boundaries of SV region for this SV event are determined where density drops to 10% of the peak summit height. All raw signals located within the SV region are then merged into one SV candidate and then passed to multi-allelic SV classification. We have updated the Methods section for SV raw signal detection and clustering.

Figure S2 SV Density-based SV raw signal clustering. **a** SV raw signals from the same chromosome with the same SV type are sorted based on coordinates. **b** Density of raw signals is calculated for each base pair on the reference genome. DeBreak scans through the chromosome for peaks above a defined threshold. **c** For each peak, boundaries of the SV region for a SV event are determined where density drops to 10% of the peak summit height. All raw signals located within the SV region are merged into one SV candidate.

c) It is unclear to us what happens to translocations during the SV detection and clustering phase; in particular, it is unclear to us how translocations between different chromosomes are detected / handled (as chromosomes are processed independently).

Response: Thanks for pointing this out. The translocations are detected when two alignments of the same read are mapped to two distinct chromosomes. In standard BAM file, each read alignment has an 'SA' tag, which stores the information (chromosome, location, CIGAR string, etc.) about all its supplementary alignments. When DeBreak processes reads on one chromosome, it checks all supplementary alignments of the read and see if there is any alignment on distinct chromosome that fits TRA pattern as shown in Fig. S1.

SV raw signal clustering is performed for each SV type separately. For DEL, INS, DUP, and INV, density-based clustering (Fig. S2) is used. For translocation, there is no SV size, so DeBreak uses a fixed window of 400/800bp for clustering both breakpoints of translocation raw signals. The window size is determined by standard variation of TRA breakpoint positions from raw signals, where an 800bp window is used for regions with larger standard variation and a 400bp window is used for regions with smaller standard variation. We have added the detection and clustering of TRA raw signals to the Methods section.

d) Large insertion detection via local assembly: How exactly is "enriched clipped alignment" defined? It is not clear to us how this step differs from the main raw SV signal detection step - which is also followed by a POA-based reassembly step? Also, it would seem to us that one key problem with long insertions is that the insertion may be longer than the read length - if we understand the approach of DeBreak correctly, however, the local re-assembly will only include reads that extend into the areas outside of the insertion (and thus miss the "middle region" of long insertions)? Last, but not least, how exactly is the local de novo assembly step carried out (e.g. using an external long-read assembly algorithm)?

Response: Thanks for point this out. The major difference between ‘enriched clipped alignment’ and SV raw signal detection is that raw signal for insertions requires both the left and right side of inserted sequences are mapped, and two alignments should be close to each other on the reference genome (**Fig. R3a**). However, for ‘clipped alignment’ of ultra-large insertions, we only require one side of the read to be mapped and the other side can be unmapped (**Fig. R3b**), as the read itself is not expected to cover both side of the large insertion. In this case, only one alignment is needed. When multiple reads (the threshold is also adjustable according to sequencing depth) are clipped at the same position, we believe that there is an ‘enriched’ clipped alignment here. We do need enriched clipped alignments on both side of the candidate INS breakpoint (a group of reads aligned to left side of breakpoint, and another group of reads aligned to right side), so that we can collect reads for assembly (**Fig. R3c**). In theory, the maximal size of ultra-large insertion detected by DeBreak is roughly twice of the read length, as we need reads from left side to reach reads from right side so that they can be connected with assembly. For INS even larger than twice of read length, there are reads from middle

Figure S3 Ultra-large INS detection. **a** Example of reads with clipped alignment. Enriched “clipped” reads are required for both side of the candidate INS breakpoint for following local assembly. **b** Local *de novo* assembly using “clipped” reads. Reads aligned to both sides of the candidate INS breakpoint are collected for local assembly to generate an assembly contig that includes the full-length insertion sequence.

region that cannot be captured through this approach. These middle reads are either unmapped (if inserted sequence is novel) or mapped to other region of the genome (such as mobile element insertion), and we will try to capture these reads to further improve detectable insertion size limit in future versions. Finally, local *de novo* assembly was done by wtdbg2. We have added a supplementary figure (**Fig. S3**) and updated Methods section regarding ultra-large insertion detection.

Minor:

- General points:

a) PBHoney was mentioned in the introduction but is not evaluated - could the authors explicitly comment on why this is (or alternatively add it to the evaluations)?

We would strongly recommend that you address this point by including a comparison to PBHoney if possible for further consideration at *Nature Communications*. This point could be addressed textually, for *Communications Biology*.

Response: Thanks for pointing this out. PBHoney was mentioned in the introduction section as it is one of the earliest SV callers developed for long reads. However, it was not properly maintained in recent years, especially after its dependencies have been updated. We failed to install and run PBHoney after many

attempts, and therefore could not include PBHoney for comparison. PBHoney was also mentioned in cuteSV's paper (Jiang et. al, 2020) but not tested for SV discovery performance, possibly due to the same reason.

- Simulations:

a) There are 3 simulated datasets with varying read lengths, but Table 1 lumps them together - it would be good to see results broken down by read length (e.g. in the Supplement) to better understand potential impacts of read length. How exactly were translocations modeled (inter-chromosomal or across chromosomes)?

Response: Thanks for the suggestion. We have added a supplementary table to list SV discovery accuracy (F1 score) in three replicates for each type of SV. DeBreak achieved the highest F1 score in most SV types using both PacBio and Nanopore data (**Table S1**). In general, the performance of DeBreak was not largely affected by read length, as the three replicates differed by <1% in most groups. We have updated the Results section and supplementary tables to help with interpretation of the results.

Table S1 SV discovery accuracy (F1 score) on three replicated simulated datasets

Type	DeBreak			Sniffles			pbsv			cuteSV		
	Rep1	Rep2	Rep3	Rep1	Rep2	Rep3	Rep1	Rep2	Rep3	Rep1	Rep2	Rep3
PacBio												
DEL	99.54	99.53	99.56	97.56	97.60	97.69	98.07	98.49	98.59	98.19	98.40	98.48
INS	98.89	99.09	99.16	94.95	96.25	96.20	95.58	96.13	95.69	93.96	94.96	95.01
DUP	98.40	98.49	98.44	94.11	94.90	94.63	64.52	62.06	56.33	60.54	61.91	60.71
INV	99.65	99.05	99.85	95.79	95.68	96.16	89.20	91.01	91.21	65.75	65.82	66.06
TRA	99.50	99.75	98.99	96.55	96.77	96.53	39.63	39.96	40.00	66.67	66.78	66.67
Total	99.20	99.26	99.34	96.15	96.78	96.81	94.12	94.58	94.32	92.53	93.12	93.16
Nanopore												
DEL	98.52	98.38	98.45	96.83	98.38	98.45	98.74	98.59	98.47	98.12	98.04	98.06
INS	99.03	98.85	98.88	96.02	98.85	98.88	96.63	96.45	96.36	94.97	95.13	94.94
DUP	95.00	95.39	95.42	94.89	95.39	95.42	62.07	57.37	61.25	55.72	57.43	57.00
INV	94.99	94.57	95.55	95.01	94.57	95.55	93.06	94.29	93.79	66.03	65.99	66.19
TRA	94.18	98.48	95.83	95.15	98.48	95.83	39.92	40.04	40.50	66.67	66.67	66.55
Total	98.40	98.29	98.35	96.28	98.29	98.35	94.97	94.77	94.78	92.89	92.97	92.89

The unit for the F1 score is %. The highest F1 score in each replicate is shown in bold. Rep1, replicate 1 (10kbp). Rep2, replicate 2 (15kbp). Rep3, replicate 3 (20kbp).

All translocations in the simulated datasets were inter-chromosomal. For each translocation event, we

simulated two breakpoints on two distinct chromosomes, and then connected sequences from two chromosomes to mimic translocation. For example, in **Fig. R4**, we simulated two translocations with random breakpoints on three chromosomes. For each translocation event, we connected sequences on the left side of breakpoint 1 and right side of breakpoint 2 as one new chromosome, and we connected sequences on right side of breakpoint 1 and left side of breakpoint 2 into the other new chromosome. After processing all translocation events, a new set of chromosomes will be generated. We have updated the Methods section to clarify the simulation dataset generation.

b) Adding a brief explanation where the different SV length simulation peaks in Figure S1 come from to the figure's legend would be helpful.

Response: Thanks for the suggestion. We have added "Peaks at 300-350bp were simulated to mimic Alu elements, and peaks near 6kbp were simulated to mimic LINE mobile elements." to the figure legend of Figure S1.

c) Figure S3b seems to show a systematic bias with respect to the breakpoint accuracy of DeBreak-determined duplications - could the authors comment on that?

Response: Thanks for pointing this out. In the simulated datasets, we simulated 1000 tandem duplications with various sizes (**Fig. R5a**). For large duplications, sequencing reads containing duplicated sequences (green section) are split into two alignments, and we can often infer the correct breakpoint locations (**Fig. R5b**).

Figure R4 Simulation of translocations. Number of SV detected with exact breakpoint and with ≤ 1 bp shift in breakpoints using PacBio CLR, HiFi, and Nanopore data at different sequencing depth. PacBio CLR data was downsampled from 10x to 70x. PacBio HiFi and Nanopore data were downsampled from 10x to 100x.

These are the duplications with breakpoint shift < 5 bp in previous **Fig. S3b** (now **Fig. 7b** after revision).

For shorter duplications, the aligner usually considers them as inserted sequences and reports an insertion in the read alignment (**Fig. R5c&d**). Ideally, the insertion site should be on the leftmost point of duplicated region, and we can infer the correct breakpoint locations (**Fig. R5c**). However, in reality, the aligner sometimes opens the gap for insertion in the middle of the duplicated region, due to the presence of sequencing errors and inherent differences between two duplicated units (**Fig. R5d**). In this case, the inferred SV breakpoint (black arrow) will shift towards right side of the true breakpoint location (green arrowhead).

In order to fix the breakpoint shift problem, DeBreak implemented a breakpoint-correction module to

Figure R5 Duplication detection. **a** Example of a tandem duplication. **b** Duplications identified from split-read alignment have correct breakpoints. **c** Duplications identified from ideal within-read alignment have correct breakpoints. **d** Breakpoints of duplications identified from within-read alignment shift towards right side. **e,f** Breakpoint accuracy of duplications in simulation dataset before and after applying breakpoint correction.

infer the original leftmost breakpoint by re-aligning inserted sequences to reference sequence surrounding the current breakpoint. With this module, we can solve this problem to some extent (**Fig. R5 e,f**), where ~50% of duplications had shift of 150-250bp before correction and shift <100bp after correction, but we still cannot achieve the 'exact' breakpoint for duplications due to the noisy alignments of inserted sequences.

d) Also, the red bars in Figure S3b are kind of hard to identify (apart from the outliers, see previous point) – consider modifying the graph to enable a better distinction between the variant types?

Response: Thanks for the suggestion. We have modified **Figure S3b** by breaking the y-axis into two segments, with the bottom part representing 0-1,000 and the top part representing 1,000-15,000. The duplications (red) and inversions (purple) are now more distinguishable now.

Figure S7 SV breakpoint refinement in simulated datasets. **a** The identity of reads/consensus sequences compared with sequences around simulated SVs for individual SV type. **b** Shift of SV breakpoints inferred from raw reads (left) and from consensus sequences (right) in three simulated datasets (top, center, and bottom). SVs with shifts more than 20bp were combined into the first and last bins.

e) The simulations could be extended by a simulated Nanopore dataset; or a discussion point could be added explaining why the results of a Nanopore simulation would be expected to be similar to the PacBio simulation results.

Response: Thanks for the advice. We have added simulation of Nanopore datasets using Badread with average read length of 10kbp, 15kbp, and 20kbp, similar to the PacBio data simulation (**Fig. S4b**). After benchmarking the performance of four SV callers, we found that DeBreak achieved the highest F1 score in detecting insertions, duplications, inversions, and translocations, while cuteSV achieved a higher F1 score for detecting deletions (**Table 1**). When combining all SV types together, DeBreak achieved F1 score of 99.23% on simulated PacBio data and 98.35% on simulated Nanopore data, which was higher than the other three SV callers. We have updated the Results and Methods section for benchmark in Nanopore simulation.

Figure S4 Characteristics of simulated datasets. **a** Size distribution of simulated SVs. Peaks at 300-350bp were simulated to mimic Alu elements, and peaks near 6kbp were simulated to mimic LINE mobile elements. **b** Length distributions of simulated PacBio (left) and Nanopore (right) reads in three simulated datasets.

Table 1 SV discovery accuracy on simulated datasets

Type	DeBreak			Sniffles			pbsv			cuteSV		
	Rec	Pre	F1	Rec	Pre	F1	Rec	Pre	F1	Rec	Pre	F1
PacBio												
DEL	99.59	99.50	99.54	95.50	99.83	97.62	97.21	99.58	98.38	96.94	99.82	98.36
INS	98.51	99.65	99.08	92.35	99.52	95.80	95.38	96.22	95.80	94.58	94.71	94.64
DUP	98.27	97.04	97.65	90.80	98.63	94.55	44.53	97.11	60.97	44.13	99.03	61.05
INV	99.10	99.40	99.25	94.57	97.22	95.88	82.67	99.92	90.47	96.53	50.00	65.88
TRA	99.17	99.67	99.41	97.67	95.60	96.62	97.67	25.04	39.86	99.00	50.30	66.70
Total	99.02	99.45	99.23	93.85	99.47	96.58	93.36	95.34	94.34	93.50	92.39	92.94
Nanopore												
DEL	98.08	98.82	98.45	94.83	98.72	96.74	98.65	98.55	98.60	97.87	98.27	98.07
INS	98.17	99.68	98.92	92.70	99.73	96.09	97.05	95.92	96.48	96.04	94.01	95.01
DUP	97.00	93.60	95.27	91.97	98.37	95.06	46.03	87.37	60.23	39.67	99.50	56.72
INV	91.63	98.71	95.04	93.67	95.18	94.41	88.20	99.96	93.71	97.37	50.00	66.07
TRA	92.83	99.83	96.17	97.17	91.38	94.18	97.83	25.26	40.15	99.17	50.17	66.63
Total	97.73	98.96	98.35	93.71	98.91	96.24	95.08	94.60	94.84	94.41	91.46	92.91

The unit for recall, precision, and F1 score is %. The highest recall, precision, and F1 score among four tested SV callers are marked in bold.
Rec = Recall. Pre = Precision. F1 = F1 score.

f) Line 118, "81.33% of SVs with ± 1 bp shift around the SV breakpoint": This statistic include correctly determind (= 0bp shift) breakpoints, correct? Phrasing this as "within 1bp of the true SV breakpoint" may be more clear.

Response: Thanks for the question. 81.33% does include SVs with exact breakpoint (with 0bp shift). We have revised this sentence as suggested by the reviewer to clarify the results.

- HG002 results:

a) It is not exactly clear to us how genotype accuracy is calculated here. What are the possible "genotypes" in the truth set - just 0, 1 or 2, or does the truth also include copy-number-variant genotypes? Also, is genotyping accuracy calculated on the respective algorithm's call set, or on the complete call set (both ways to evaluate genotyping accuracy are important and should be included).

Response: Thanks for raising this concern. The genotype of SVs in the ground-truth callset includes '0/1' (heterozygous) and '1/1' (homozygous). Copy number variants are not included in the ground-truth callset. SV callers also report SVs as heterozygous ('0/1') and homozygous ('1/1'). We compared the genotypes reported by each SV callers and considered an SV to be correctly genotyped if the genotype tag was the same with ground truth.

Genotyping accuracy was calculated based on the algorithm's callset, with the number of correctly genotypes SVs divided by the number of total SVs reported by each SV caller. We assessed genotyping accuracy for each SV caller to reflect its ability to correctly predict genotype for the SVs it has identified. The evaluation was not done based on the complete callset, as some false-negative SVs will be falsely considered as 'incorrect genotype'. These false-negative SVs should be considered when evaluating the recall of SV callers instead of genotyping accuracy.

b) Please also specify +/-1bp breakpoint accuracy for all algorithms in the text (to make results fully comparable to the simulation experiment).

Response: Thanks for the advice. We have added the percentage of SVs reported with breakpoint shift within 1bp for four SV callers in the manuscript. In PacBio CLR dataset, DeBreak identified 59.90% of SVs with exact SV breakpoints and 63.53% of SVs with breakpoint shift within 1bp as reported in GIAB, which was higher than pbsv (41.73% and 47.41%), Sniffles (4.99% and 13.45%), and cuteSV (5.18% and 13.87%).

c) Figure 2: Please make the utilized read data type (CLR, HiFi, Nanopore) more explicit in the figure and the legend.

Response: Thanks for the suggestion. We have labeled the data type used in each plot in their subtitles and also specified datasets used in the figure legends for **Figure 2**.

Figure 2 SV discovery in HG002. **a** SV discovery accuracy for insertions (positive SV size) and deletions (negative SV size) at different size ranges in CLR dataset. Bars indicates the number of SVs in each size range, and lines show the SV discovery accuracy for each SV caller. **b** SV breakpoint accuracy for four tested SV callers in CLR dataset. SVs with breakpoint shifting >100bp were included in the ± 100 bp bins. **c** SV discovery accuracy in downsampled PacBio CLR (left), HiFi (middle), and Nanopore (right) datasets.

- HGSVC results:

a) Please also specify ± 1 bp breakpoint accuracy for all algorithms in the text (to make results fully comparable to the simulation experiment), and specify breakpoint accuracy as % of calls made

Response: Thanks for the suggestion. We have replaced the exact number of SVs to percentage of SVs. We have also added the percentage of SVs identified within 1bp shift of true breakpoints for the four SV callers: “With the breakpoint-refinement module, DeBreak identified 46.83% of SVs with exact SV breakpoints and 48.12% of SVs within 1bp shift on the three PacBio CLR datasets, while pbsv reported 39.03% and 40.99%, Sniffles reported 3.02% and 6.82%, and cuteSV reported 2.75% and 7.69% of SVs with exact breakpoints and within 1bp shift, respectively (Fig. 3b). For the three PacBio HiFi datasets, DeBreak also achieved better breakpoint accuracy than the other three callers, as 56.98%, 47.41%, 2.99%, and 15.33% of SVs were identified with exact SV breakpoints, and 58.09%, 48.85%, 6.41%, and 35.54% of SVs were identified within 1bp shift by DeBreak, pbsv, Sniffles, and cuteSV, respectively.”

b) It would be interesting (though not essential) to see results for multi-allelic copy number variations here as well.

Response: Thanks for the advice. We selected the multi-allelic SVs in the HGSVC samples for further analysis. As all six samples are healthy individuals, there were similar number of mSVs in each sample (Table S7). A total of 3,100 mSVs (6,200 alternative alleles) were detected in three CLR datasets, and 3,097 mSVs (6,194 alternative alleles) were identified in three HiFi datasets. We then compared these alternative alleles to the assembly-based PAV callset and validated 71.98% and 71.47% of them in CLR and HiFi datasets, respectively.

Table S7 DeBreak mSV discovery in HGSVC samples

	mSV	Alternative allele	Validation rate	mCNV
CLR				
HG00096	1097	2194	73.38	12
HG01505	1011	2022	71.96	4
HG01596	992	1984	70.46	7
Total	3100	6200	71.98	23
HiFi				
HG02818	1031	2062	73.81	7
HG03486	1187	2374	72.11	10
NA12878	879	1758	67.86	7
Total	3097	6194	71.47	24

Alternative alleles that are also reported in assembly-based SV callset were considered as validated. mCNV is classified using k-mer counts. The unit of validation rate is %.

For these mSVs, we further annotated multi-allelic copy number variations using a k-mer approach. For each alternative allele, we extracted the sequences (deleted reference sequences for deletion; inserted sequences for insertions) and counted the k-mer occurrence with $k=15$. We consider an alternative allele to be copy number variation if at least 70% of the k-mer have occurrence larger than 1, so that there are more than 1 copy of a certain segment. An mSVs was considered as mCNV if both its alternative alleles were considered as CNV. For example, we extracted the SV sequence and the flanking sequences from reference genome to count the occurrence of all unique k-mers for an mSV in HG02818. Based on the peak in the histograms, there are 9 copies in the reference genome (**Fig. S20a**), 11 copies in alternative allele 1, and 14 copies in alternative allele 2 (**Fig. S20b,c**). Alternative allele 1 was an insertion of 94bp and added two copies. Alternative allele 2 was an insertion of 235bp and added five copies, which suggests a consistent repeat unit size (47bp) with alternative allele 1. The other k-mers with lower occurrence may be k-mers containing sequencing errors. We further validated the identity of sequences by aligning the inserted sequences of two insertion alleles back to the reference genome, and the sequences of both alleles can be fully mapped back to reference (**Fig. S20d**). Using this k-mer approach, we identified 23 mCNVs in the three CLR datasets and 24 mCNVs in three HiFi datasets (**Table S7**). We have added the supplementary figure/table and updated the Results section regarding the mSV analysis in HGSVC samples.

Figure S20 Example an mCNV event (chr12-1079285-94bp-INS/chr12-1079285-235bp-INS) in HG02818. **a** k-mer occurrence of reference sequences. A peak of k-mer at 9 indicates 9 copies of a specific repeat unit in the reference genome. **b, c** k-mer occurrence of alternative allele 1 (**b**, INS of 94bp) and alternative allele 2 (**c**, INS of 235bp). 11 and 14 copies of the repeat unit are present in the two alleles. **d** Alignment of inserted sequences of two alternative alleles. Both alleles can be fully aligned to the reference genome near the mSV breakpoint.

c) Line 207, "suggesting that alignment-based SV callers identified SV calls more consistently.": More consistently than assembly-based methods? As only one assembly-based data source (PAV) is considered here, this claim does not seem to follow.

Response: Thanks for pointing this out. We intended to describe that the SV callset of one alignment-based caller is more similar with another alignment-based caller than with an assembly-based caller. We have removed this statement, as it could cause confusion for readers, and we agree that the conclusion was not strongly supported since we only compared to one assembly-based approach.

- Cancer genome results:

a) A brief description of the input data (technology type, coverage, read lengths) in the main text would be very helpful.

Response: Thanks for the advice. We have added the datatype, coverage, and average read length of SKBR3 dataset in the results section: "PacBio CLR data (72x, mean read length of 9.87kbp) of SKBR3 was downloaded and aligned to human reference genome." The Data Availability contains the access number of this dataset.

b) Would it be possible to construct an assembly-based "truth set" here as well?

Response: Thanks for the suggestion. Unfortunately, we could not generate a high-confidence truth set of SVs from assembly with the data available (PacBio CLR and Iso-Seq). The six HGSC samples were sequenced by multiple platforms (PacBio, Strand-Seq, Bionano, etc). Haplotype-resolved assembly was performed with a combination of input data types. SV callsets were called from the high-quality assemblies and then manually curated by the HGSC group (Ebert et. al., 2021), which ensured their high confidence to serve as ground-truth callsets. In SKBR3, we could not generate such high-quality haplotype-resolved assembly with only CLR data, which means we cannot obtain high-confidence/true SV callset from the assembly.

c) The fusion gene-specific validation approach leveraging IsoSeq data is very interesting. Could the same approach be applied to the results of the other SV calling methods (to the extent that they are capable of calling translocations) as well? (We would view this as very interesting, though not essential)

Response: Thanks for the suggestion. We have annotated SVs reported by Sniffles, pbsv, and cuteSV to identify potential gene fusion calls. 41 gene fusions were reported by DeBreak, and 11 of them were cross-validated in Iso-Seq data. Sniffles reported 116 gene fusions, and 25 of them were cross-validated. pbsv reported 39 gene fusions, and 5 of them were cross-validated. cuteSV reported 58 gene fusions, and 7 of them were cross-validated. We have added the cross-validation results for the other three SV callers to the Results section.

- Methods:

a) Explicitly mentioning the multi-allele step of the workflow may be helpful.

b) Line 312: "The density of SV raw signal is computed for each position on the chromosome (Supp fig)" -> Figure number missing.

c) Section 3.33 only mentions the PacBio CLR dataset, omitting the Nanopore and HiFi datasets (which are also downsampled).

d) Line 342: Please define when exactly a read is counted as "supporting"

Response: We appreciate the reviewer's careful proofreading of the manuscript.

- a) We have added multi-allele SV identification to the Methods section 1.1 overall workflow of DeBreak. Detailed methods of mSV detection are introduced in Methods section 1.2.

b) We have removed the "(supp fig)" in line 312, as no supplementary figure should be linked here.

c) We have added PacBio HiFi and Nanopore data downsampling to Methods section 3.3.

A read is counted as SV-supporting read if it contains raw signal of a particular SV event. We have added the definition of SV-supporting read in Methods section 1.4.

Reviewer comments:

Reviewer #1 (Remarks to the Author: Strength of the claims):

The authors have adequately addressed most of my comments. I have a couple of follow-up questions on 3) and 6) in my initial review.

1) I am surprised by the lower recall after applying dipcall confident regions. Also, the low recall on CHM13 remains surprising. The authors may have a look at Figure 4B in the HPRC paper (doi.org/10.1101/2022.07.09.499321): both recall and precision are above 90%. That subfigure still compares assembly based calls, but a recall below 80% in the current paper needs an explanation. I recommend the authors to pick some FNs/FPs and manually inspect in IGV. They will probably see many complex cases especially around VNTRs.

I also recommend the authors to try truvari. SV comparison is challenging. I worry the authors' script may have overlooked complications around VNTRs. When running truvari, it is important to apply "--multimatch" and a large "-r" (e.g. 1000). Some people use even more relaxed settings such as "--multimatch -r 1000 -C 1000 -O 0.0 -p 0.0 -P 0.3 -s 50 -S 15 --sizemax 100000 --includebed", though I don't know if that is necessary.

The authors' evaluation script gives meaningful results on HG002 possibly because assembly-based SVs are enriched with VNTRs and are harder to evaluate (see Fig 4 in PMID:33066802).

Minor comment

2) It would be good to make the evaluated callsets available. I was thinking to try truvari by myself, but I couldn't find the download link to authors' calls.

3) The definition of mSV is tricky. For example, suppose at a site the two insertion alleles differ by one basepair. Is this a mSV or not? What if the two alleles differ by 20bp in length? What's the authors' definition of mSV?

Reviewer #2 (Remarks to the Author: Overall significance):

The manuscript presents an alternative approach to SV detection incorporating local reassembly, a nice approach to better resolve SVs.

Reviewer #2 (Remarks to the Author: Impact):

The new method of resolving SVs (or at least, incorporating local reassembly) is likely to improve the range of SV callers that exist.

Reviewer #2 (Remarks to the Author: Strength of the claims):

In general all comments I raised on the previous review have been addressed.

The authors have moderated the claims on SVs - I think the evaluation of significance by bases affected is

misleading - the text now in the paper is more appropriate.

Similarly claims on performance are accurately reported.

Reviewer #2 (Remarks to the Author: Reproducibility):

New conda install is effective and the update to python3 is good. Switching to use pySam will improve writing of VCF files.

I have not had time to test the code on real samples.

Reviewers #3-4 (Remarks to the Author: Overall significance):

We would like to thank the authors for carefully addressing the points we raised.

There are two remaining minor points:

1- In line 615, the Version number of Debreak is mentioned to be v1.0.2, while in the method section it's mentioned to be v1.2.

2- The method section is definitely well explained. This makes us raise this question: If two types of variations exist at the same position, will Debreak be able to detect both types? For example, if a translocation was accompanied by a duplication event (a complex structural variation event)?

Apart from that, we have no further comments, and we now recommend the paper for publication.

Reviewer #1 (Remarks to the Author: Strength of the claims):

The authors have adequately addressed most of my comments. I have a couple of follow-up questions on 3) and 6) in my initial review.

1) I am surprised by the lower recall after applying dipcall confident regions. Also, the low recall on CHM13 remains surprising. The authors may have a look at Figure 4B in the HPRC paper (doi.org/10.1101/2022.07.09.499321): both recall and precision are above 90%. That subfigure still compares assembly based calls, but a recall below 80% in the current paper needs an explanation. I recommend the authors to pick some FNs/FPs and manually inspect in IGV. They will probably see many complex cases especially around VNTRs.

I also recommend the authors to try truvari. SV comparison is challenging. I worry the authors' script may have overlooked complications around VNTRs. When running truvari, it is important to apply "--multimatch" and a large "-r" (e.g. 1000). Some people use even more relaxed settings such as "--multimatch -r 1000 -C 1000 -O 0.0 -p 0.0 -P 0.3 -s 50 -S 15 --sizemax 100000 --includebed", though I don't know if that is necessary.

The authors' evaluation script gives meaningful results on HG002 possibly because assembly-based SVs are enriched with VNTRs and are harder to evaluate (see Fig 4 in PMID:33066802).

Response: We thank the reviewer's comments and suggestions regarding the comparison of dipcall results. In the manuscript, we used the same criteria for all benchmark: two SVs are considered as matched when they have the same SV type, the breakpoints are located within 1kbp, and the size ratio is larger than 0.5. Multimatch was not allowed in our custom script.

We have tested Truvari to compare four alignment-based SV callers to Dipcall callsets in HGSVC samples with options "-r 1000 --pctsim 0.0 --multimatch -P 0.5 --passonly". (--pctsim 0.0 was used to avoid comparing allele sequences. -P 0.5 and --passonly were used to keep it consistent with our benchmark.) Overall, the recall reported by Truvari (~68-84%) was ~11% higher than our custom script (~59-73%) (**Figure R1A,C**). We further investigated the differences

and found two major reasons. First, our custom script does not allow multimatch. When enabling multimatch, we observed ~5% improvement in recall for all six samples. Second, Truvari set a minimal SV size of 50bp for Dipcall callset, while we used a cutoff of 45bp allowing some shifts. Under the setting of the minimal SV size of 50bp and allowing for multimatch, the recall reported by our custom script was similar to the benchmark results of Truvari, with a recall of 81-83% in both whole genome and confident regions for DeBreak (**Figure R1B,C**).

Figure R1 Alignment-based SV callsets compared to Dipcall callsets. Recall and Precision evaluated with custom script in manuscript (**A**), custom script allowing multimatch (**B**), and Truvari (**C**) in whole genome (left) and Dipcall confident regions (right).

We have also tested the relaxed settings of Truvari, “-r 1000 --pctsim 0.0 --multimatch -C 1000 -O 0.0 -p 0.0 -P 0.3 -s 50 -S 15 --sizemax 100000 --passonly”, as suggested by the reviewer. (We added `--passonly` because cuteSV callsets included many SVs with other flags and would have low precision when including all SVs.) Using the more relaxed settings, the recall was further increased by ~3% for all SV callers, with a recall of 84-86% for DeBreak. The F1 score reached 87-90% for DeBreak and cuteSV in confident regions (**Table R1**), which was consistent with the HPRC preprint paper the reviewer referred to.

Table R1. SV discovery accuracy (F1 score) under relaxed settings

Sample	Whole genome				Confident regions			
	DeBreak	sniffles	cuteSV	pbsv	DeBreak	sniffles	cuteSV	pbsv
HG00096	81.62	80.04	84.08	79.27	87.51	83.48	90.20	85.88
HG01505	81.93	79.55	83.93	78.87	87.64	82.87	90.05	85.47
HG01596	79.55	77.84	80.37	78.28	85.83	81.32	87.23	85.24
HG02818	83.84	79.54	83.57	78.08	88.87	82.55	88.30	83.41
HG03486	84.30	80.75	84.64	78.48	89.49	83.72	89.56	83.74
NA12878	84.09	77.89	82.76	77.99	89.42	80.88	87.16	82.60

Highest F1 score among four tested SV callers was marked as bold for each sample.

Dipcall was only applied during Revision version 1 on CHM13. As shown in the revised **Table S8**, we have now updated the benchmark methods used in CHM13 datasets by setting the minimal SV size of 50bp and allowing multimatch. DeBreak and cuteSV showed recall of 82-88% for deletion detection and 79-86% for insertion detection, while Sniffles and pbsv had lower recall. We have also updated the Methods section accordingly.

Table S8 SV discovery accuracy in CHM13

	Deletion			Insertion			Total		
	Recall	Precision	F1	Recall	Precision	F1	Recall	Precision	F1
CLR									
DeBreak	82.41	88.58	85.38	79.29	87.19	83.05	80.45	87.72	83.93
Sniffles	78.66	86.01	82.17	63.03	90.42	74.28	68.86	88.49	77.45
pbsv	81.84	84.76	83.27	70.74	85.52	77.43	74.88	85.21	79.71
cuteSV	83.84	86.35	85.08	81.44	84.74	83.06	82.33	85.35	83.81
HiFi									
DeBreak	83.71	86.12	84.90	80.81	89.89	85.11	81.89	88.39	85.02
Sniffles	79.44	83.84	81.58	60.42	88.75	71.89	67.51	86.50	75.84
pbsv	80.69	83.18	81.92	65.04	88.26	74.89	70.88	86.01	77.71
cuteSV	85.93	79.68	82.69	84.03	85.07	84.55	84.74	82.97	83.84
Nanopore									
DeBreak	85.72	82.83	84.25	83.46	87.70	85.53	84.30	85.76	85.03
Sniffles	81.87	78.57	80.19	64.49	88.54	74.63	70.97	83.94	76.91
pbsv	84.44	54.55	66.28	68.12	86.07	76.05	74.21	69.06	71.54
cuteSV	88.52	71.71	79.23	86.09	83.24	84.64	87.00	78.41	82.48

We then tested Truvari with options “-r 1000 --pctsim 0.0 --multimatch -P 0.5 --passonly” on comparison to PAV callsets of HGSC samples. The recall of four SV callers ranged from 66% to 83%, which was only slightly higher than the recall observed using custom script (64-81%) (**Figure R2**). As the PAV callset has already been merged and restricted to ≥50bp SVs, the difference between our custom script and Truvari was marginal. Therefore, we decided to keep the benchmark methods for PAV callsets in HGSC samples.

Figure R2 Alignment-based SV callsets compared to PAV callsets. Recall and Precision evaluated with custom script in manuscript (A) and Truvari (B).

Minor comment

2) It would be good to make the evaluated callsets available. I was thinking to try truvari by myself, but I couldn't find the download link to authors' calls.

Response: We have made the VCF files used in the manuscript available at Zenodo: <https://zenodo.org/record/7214225>.

3) The definition of mSV is tricky. For example, suppose at a site the two insertions alleles differ by one basepair. Is this a mSV or not? What if the two alleles differ by 20bp in length? What's the authors' definition of mSV?

Response: Multi-allelic SVs (mSVs) are defined as two segregating alleles in a region with different SV types (i.e., DEL and DUP) or with same SV type and the SV sizes differ by at least 50% (e.g., a 100bp INS and a 150bp INS are considered as mSVs). Alleles differ by one or two basepairs were not reported as mSVs, because such small differences can easily be caused by sequencing errors. Two alleles differ by 20bp can be tricky. 20bp can be a minor difference for SVs over 1kbp, which could be caused by alignment ambiguity in VNTRs or sequencing errors. Under this circumstance, the two alleles are not reported as mSVs based on our definition. However, 20-bp difference can be quite significant for a small SV, for example a 50bp SV. So, we used relative ratio instead of absolute value to determine whether there are two different alleles (mSV) or just one allele.

Reviewer #2 (Remarks to the Author: Overall significance):

The manuscript presents an alternative approach to SV detection incorporating local reassembly, a nice approach to better resolve SVs.

Reviewer #2 (Remarks to the Author: Impact):

The new method of resolving SVs (or at least, incorporating local reassembly) is likely to improve the range of SV callers that exist.

Reviewer #2 (Remarks to the Author: Strength of the claims):

In general all comments I raised on the previous review have been addressed.

The authors have moderated the claims on SVs - I think the evaluation of significance by bases affected is misleading - the text now in the paper is more appropriate.

Similarly claims on performance are accurately reported.

Reviewer #2 (Remarks to the Author: Reproducibility):

New conda install is effective and the update to python3 is good. Switching to use pySam will improve writing of VCF files.

I have not had time to test the code on real samples.

Response: Again, we are grateful for the reviewer's positive comments, which were very helpful for the improvement of our tool and the manuscript.

Reviewers #3-4 (Remarks to the Author: Overall significance):

We would like to thank the authors for carefully addressing the points we raised.

Response: We also would like to thank the reviewers' effort in evaluating our manuscript. All the comments are very helpful.

There are two remaining minor points:

1- In line 615, the Version number of Debreak is mentioned to be v1.0.2, while in the method section it's mentioned to be v1.2.

Response: Thanks for pointing this out. The version of DeBreak used in the manuscript is v1.2. We have corrected the typo in the Code availability section to v1.2.

2- The method section is definitely well explained. This makes us raise this question: If two types of variations exist at the same position, will Debreak be able to detect both types? For example, if a translocation was accompanied by a duplication event (a complex structural variation event)?

Apart from that, we have no further comments, and we now recommend the paper for publication.

Response: DeBreak can detect multiple types of variations at the same location. During the SV raw signal detection and clustering processes, each SV type is processed separately, so the presence of one SV type should not affect the detection of other types at the same region. We were able to find a few examples in HG00096, as shown in **Figure R3**. A 1807bp deletion located within duplicated region of 4259bp was reported by DeBreak, and a 713bp insertion located within 4009bp inversion was also reported. These events can be considered as substrates for complex SV analysis.

Figure R3 Examples of complex SVs in HG00096. PacBio CLR read alignments of HG00096 at two complex SV regions. SV type, position and size are labeled near each SV call.

REVIEWERS' COMMENTS:

Reviewer #1 (Remarks to the Author: Strength of the claims):

The authors have addressed my concerns.

Reviewer #1 (Remarks to the Author: Strength of the claims):

The authors have addressed my concerns.

Response: We appreciate the reviewer's comments and suggestions along with the revisions. All the comments are very helpful to improve the manuscript!